# Urban Regeneration and Soft Mobility: The Case Study of the Rimini Canal Port in Italy

Rachele Corticelli [1,*], Margherita Pazzini [2,*], Cecilia Mazzoli [1], Claudio Lantieri [2], Annarita Ferrante [1] and Valeria Vignali [2]

1    Department of Architecture, Alma Mater Studiorum-Università di Bologna, 40126 Bologna, Italy
2    Department of Civil, Chemical, Environmental and Materials Engineering, Alma Mater Studiorum-Università di Bologna, 40126 Bologna, Italy
*    Correspondence: rachele.corticelli2@unibo.it (R.C.); margherita.pazzini2@unibo.it (M.P.)

**Abstract:** The increasing need to reduce emissions and the environmental impact of urban areas to meet European decarbonisation goals motivates the selection of the Rimini Canal Port as a case study within the FRAMESPORT project, part of the European Interreg Italy–Croatia programme. A preliminary historical–documental and urban regulations analysis of the context allowed the identification of the main criticalities and potentials through a SWOT analysis. The central role of the stakeholders enabled the creation of a successful participatory co-design process developed through online surveys. Critical issues that emerged during the data collection phase were prioritised through a BOCR model, a powerful multi-criteria analysis tool. The project phase then focused on the resolution of the two main critical issues that emerged: the improvement of cycle/pedestrian paths, and the raising of the flooding docks in the Canal Port area. This article intends to demonstrate the strong influence of soft mobility in urban regeneration projects, and how an improvement of the quality of cycle/pedestrian paths can increase the quality of urban spaces. The new paths create a green infrastructure that contributes to a reduction in pollutant emissions through the promotion of sustainable mobility systems and an increase in green urban spaces.

**Keywords:** urban regeneration; cycle/pedestrian path; soft mobility; green infrastructure; sustainable mobility; public space design

## 1. Introduction

Climate change and environmental degradation are the challenges that the European Green Deal intends to address in order to pursue the goal of reducing net greenhouse gas emissions by at least 55% by 2030 and achieving net zero emissions by 2050 [1]. In this framework, the EU Strategy on Green Infrastructure aims to outline how to develop, preserve, and enhance healthy green infrastructure to help stop the loss of biodiversity and enable ecosystems to contribute to the collective well-being [2].

The EU Interreg Italy–Croatia Cooperation Programme addresses key socioeconomic challenges related to the sustainable growth of small realities across Europe [3]. Given the richness of cultural and environmental resources that require appropriate conservation in the programme area, it is relevant to identify a strategy to develop and preserve the heritage and at the same time promote it for tourism. Since a large part of the Adriatic Sea is subject to national jurisdiction limitations, it is necessary to intervene with cross-border cooperation projects to ensure efficient protection of marine biodiversity and to make sustainable use of coastal and marine resources. The Interreg Italy–Croatia programme enhances the application of innovative methods to reduce the environmental impact in marine and coastal areas. The activities carried out in the FRAMESPORT (FRAMEwork Initiative Fostering the Sustainable Development of Adriatic Small PORTs) Interreg Italy– Croatia project [4] are aimed at supporting the overall and sustainable growth of Adriatic

Sea small ports through a long-term strategy, enhancing their socioeconomic role in the development of coastal areas. In this contribution to the research, a project pertaining to this specific domain and applied to the Rimini Canal Port was analysed and developed.

The objective behind the project was to completely regenerate an urban area by attempting to minimise both its economic and its ecological impacts. In fact, an effort was made to reduce demolition to a minimum to avoid producing waste from the demolition of large urban areas, which would have contributed to increasing emissions of greenhouse gases [5]. Instead, an attempt was made to intervene with targeted and punctual interventions to reconnect the existing urban network. Special emphasis was placed on the role of mobility, and soft mobility in particular, in the regeneration of disconnected urban networks [6].

"Soft mobility" can be defined as a "zero-impact" mobility alternative to car use that involves pedestrian, cycle, and other non-motorised means of transport [7]. It belongs to all effects within the category of sustainable mobility that aims to reduce the environmental impact of the transport sector and improve the quality of life in urban areas [8]. Soft mobility contributes to urban well-being by reducing noise and pollution levels, reducing traffic congestion, and improving road safety. This vision of mobility as a benefit to health and environment comes from the experience of Sustrans (sustainable transport), a charity founded in the UK in the 1970s by a group of cyclists and environmentalists, motivated by the opportunity to reduce over-reliance on the private car in the wake of the 1973 oil crisis. Sustrans has shown that it is possible to change people's behaviour and bring benefits in terms of health, environment, quality of life, and economic value [9]. Another interesting European model of interaction between sustainable mobility and urban regeneration is the Dutch "woonerf" model. The concept behind woonerf is the sharing of the road between different users, eliminating the physical separation of pedestrian, cyclist, and vehicle space. Pavements, cycle paths and car lanes lose their usual conformation to become one continuous ground surface. This is shaped to provide a shared space where the pedestrian is not relegated to the pavement, but can move freely, thanks to a series of measures that reduce the traffic speed of cars. In woonerf, a number of strategies are applied, such as small speed bumps, greenery, and pavement design, to force car drivers to adopt low speeds.

A particular type of route that lends itself perfectly to soft mobility is the "greenway", a route that is closed to motorised traffic and is generally intended for all other types of users: pedestrians, cyclists, horse-riders, skaters, etc. It can be found both outside and inside urban centres. The first historical theorist of greenways was Frederick Law Olmsted, a landscape architect in the second half of the XIX century, but the most pertinent definition is the one provided by Tom Turner, who identifies it as an environmentally pleasant route [10]. Starting with the spread of greenways during the course of the last century, there has been a growing desire to create a network of "green corridors", in Italy called "corridoi verdi", to allow people to access public green spaces close to where they live and to connect rural and urban areas through routes dedicated to soft mobility. The design of these spaces is not trivial, because it needs to provide solutions to make the paths accessible and safe. The network of routes that composes a greenway must meet specific requirements such as width, type of paving, etc. that depend on local regulatory standards of public space design. In Italy, these characteristics are defined by the Italian Greenways Association (IGA). Here, the territory with its river parks, canal system, networks of disused railway tracks and network of rural roads and paths on the plains and mountains, embedded in a context of unique historical–cultural and agricultural–forest values, represents an ideal scenario for a design and planning development related to the greenways concept.

Even though the promotion of soft mobility emerged as a way to reduce the use of private cars for short-distance routes, at the beginning with the sole aim of reducing emissions from vehicular traffic, it has also proven to be an effective means of actively contributing to the urban regeneration of degraded areas [11].

Soft mobility has also demonstrated its potential to create a synergetic alliance with the tourism sector. In the pilot project carried out in the framework of the European project "Alpine Space" EU Interreg III, it was analysed how the implementation of public

transport by optimisation of the combination of rail, bus, taxi, bicycle, and other means of transport proved to be a winning choice for promoting more sustainable tourism with a lower environmental impact [12].

Proven the benefits that soft mobility can bring in terms of:

- Reduced pollutant emissions;
- Increased active mobility of people;
- Increased use of public spaces;
- Reduction of urban and social degradation phenomena;
- Increased local tourism.

The objective of this contribution is to demonstrate how soft mobility plays a fundamental role in a deep urban regeneration framework in a consolidated historical context, such as Rimini's Canal Port. In fact, it is intended to demonstrate that through targeted interventions to reconnect cycle and pedestrian routes, it is possible to significantly improve the quality of life in an urban area subject to degradation and currently underutilised [13]. Interventions to reconnect cycle paths and implement pedestrian accesses to the quays represent works with a low economic and environmental impact for the municipality, but which can strongly contribute to the urban transformation to improve the quality of life of inhabitants and tourists [14].

The methodology proposed within this contribution aims to identify the optimal urban regeneration solutions for the Rimini Canal Port area through an intervention strategy that reflects the sustainability criteria promoted by the Interreg Europe programme. The multidisciplinary character of sustainability follows the principles of the World Summit on Sustainable Development (WSSD) in Johannesburg in 2002 [15]. This includes not only environmental protection but also economic development and social welfare: environmental quality cannot be independent of people's well-being [16]. This is why 17 goals were defined in 2015 as part of the 2030 Agenda for Sustainable Development [17], 11 of which aim to 'make cities inclusive, safe, resilient and sustainable', criteria that were considered as a basis for the research project. After an accurate analysis of the historical and urban context of Rimini's Canal Port, it was possible to identify the area's potential and criticalities, thanks also to the participation of the stakeholders involved in the decision-making process [18]. In order to guarantee optimal use of resources and a better outcome for the final project, it was necessary to identify a method that would allow the identification of those criticalities that emerged that needed priority intervention. This was done through a Benefits, Opportunities, Costs, Risks (BOCR) model [19]. The outcomes of this analysis constituted the basis on which to set the focus of the project proposal, which was centred on an improvement of the cycle/pedestrian routes near the docks of a specific stretch of the Canal Port, particularly needful for urban regeneration intervention [20].

## 2. Data Collection

The Canal Port of Rimini was selected as a pilot case for the European Interreg Italy–Croatia FRAMESPORT project [4], whose activities are aimed at supporting the overall and sustainable growth of Adriatic Sea small ports through a long-term strategy, enhancing their socioeconomic role for the development of coastal areas. The involvement of local and national stakeholders is a key element of the project strategy [21]. With this purpose, the ITL Foundation coordinated the activities envisaged in WP5 "Innovative services and tools to support the strategic development of small ports" in collaboration with the University of Bologna.

### 2.1. Case Study Description and Analysis

In order to properly set up the design phase, it was necessary to carry out an appropriate analysis of the urban context in which the Rimini Canal Port is located. The analysis was performed by taking into consideration multiple aspects related to the area under study [22]. In a first phase, the urban, territorial, and landscape system of the Canal Port was analysed.

Rimini's Canal Port consists of the original mouth of the Marecchia River, with quays on two sides and an extension on two piers. The canal is 46 m wide at the entrance to the mouth of the port, and 40 m wide along its length up to Parco XXV Aprile and 2.2 km long. It divides the historic city centre from the district of San Giuliano a Mare in the north of Rimini. On the left of the port are activities closely related to fishing: from shipyards to machine shops, from the wholesale fish market to nautical shops. On the right of the port stands the lighthouse, the symbol of seafaring.

To provide an exhaustive and detailed description, the Canal Port was divided into eight macro-areas (Figure 1): Entrance; Largo Boscovich; Marina; Beach area; Left quay; Right quay; Historic area and Park area. The analysis of the urban, territorial, and landscape system was carried out through the collection of data obtained from inspections and surveys integrated with archive sources relating to historical–documentary information concerning each macro-area analysed.

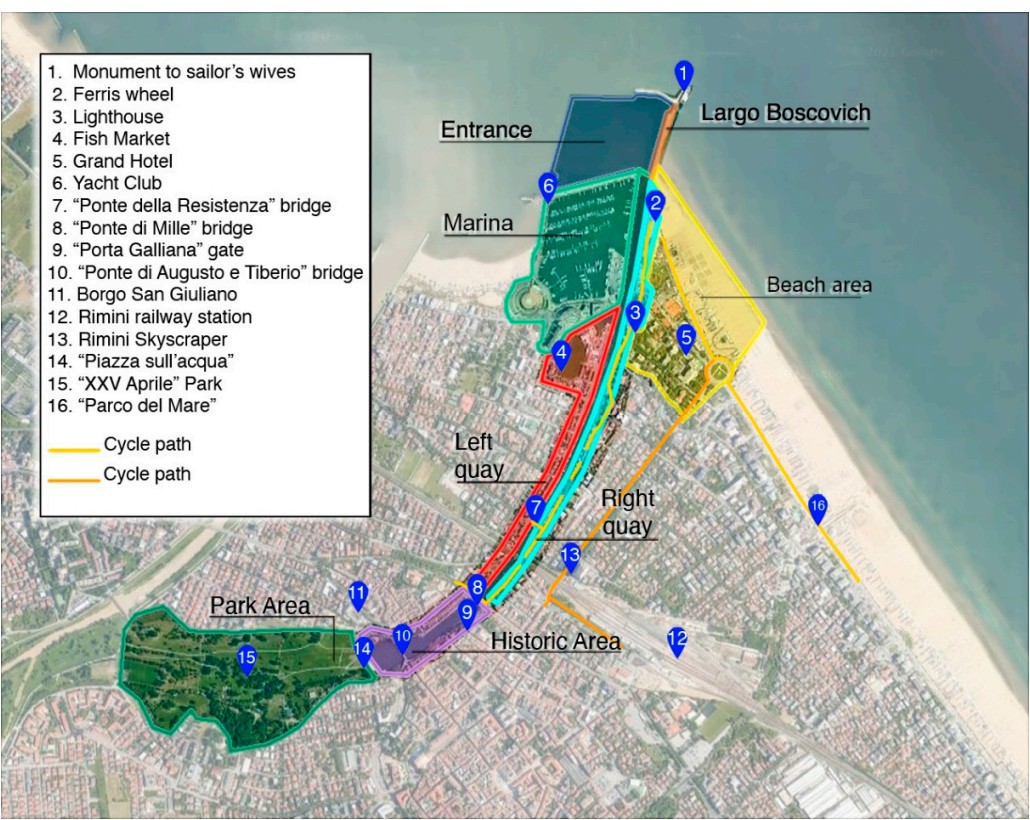

**Figure 1.** Map of the macro-areas identified during the analysis. The numbers indicate points of historical and documentary interest (© 2022, R. Corticelli—CIRI Building and Construction).

### 2.2. Analysis of Historic–Documentary Sources and Urban Regulations

After an in-depth analysis of the Canal Port in its current conformation and in relation to the transformations that have occurred over the course of time, which took place by means of surveys and archive research, a documentary analysis was carried out. It detected the regulatory discipline, urban planning instruments and maps with which the Municipality of Rimini is endowed. The first of these plans is the Piano Regolatore Generale (PRG), with which the municipality has regulated changes to the territory since its first approval in 1978. Subsequently, with Regional Law no. 20 of 2000, the PRG was divided into three general planning instruments: Piano Strutturale Comunale (PSC), Regolamento Urbanistico Edilizio (RUE), and Piano Operativo Comunale (POC).

On 1 January 2018, the new Regional Urban Planning Law no. 24 of 21 December 2017 "Regional discipline on the protection and use of the territory" came into force [23]. The new regional urban planning law obliges municipalities to start by 01 January 2021

the process of formation of the General Urban Plan (PUG) [24] that will replace the urban planning instruments provided by Regional Law 20/2000 (PSC-RUE-POC). According to this law, the Municipality has started the process of forming the PUG, and since the same law also prohibits the approval of new POCs, it made it possible to implement part of the PSC through operational agreements.

### 2.3. Analysis of the Mobility System

The mobility study needed to analyse a broader context than merely the Canal Port. For this reason, the Sustainable Urban Mobility Plan (SUMP) [25] was consulted, starting with the mobility infrastructure at a regional and supra-municipal level and going deeper down to micro-mobility. On the urban scale, the evolution of the municipal territory shows how Rimini has been strongly conditioned by the presence of barriers on the territory, both natural and anthropic: the Marecchia River, the railway lines, the A14 highway, and State Roads 9 and 16. Looking in more detail at the current state of soft mobility configuration, within the Municipality of Rimini there are several 30 km/h speed limit zones. They are mainly located between the seafront and the railway strip, covering an area of over 46,000,000 m². With the coming into force of the SUMP, their extension is also planned in the area of the historic centre and Borgo San Giuliano.

Rimini also offers a great variety of routes to cyclists, thanks to its network of cycle paths (more than 120 km of "Bicipolitana"), which go from the flat roads of the coast and the historic centre directly back to the hills. On the right side of the Canal Port there is a two-way cycle path on a raised pavement, which runs along the port beside the road. The cycle path is properly delimited with specific vertical and horizontal signs and red paving (Figure 2).

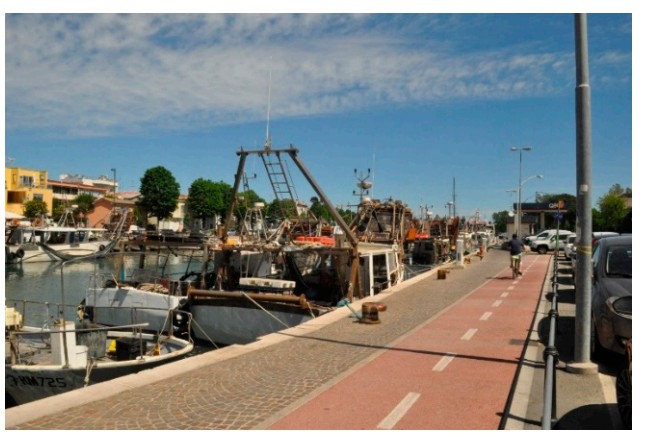
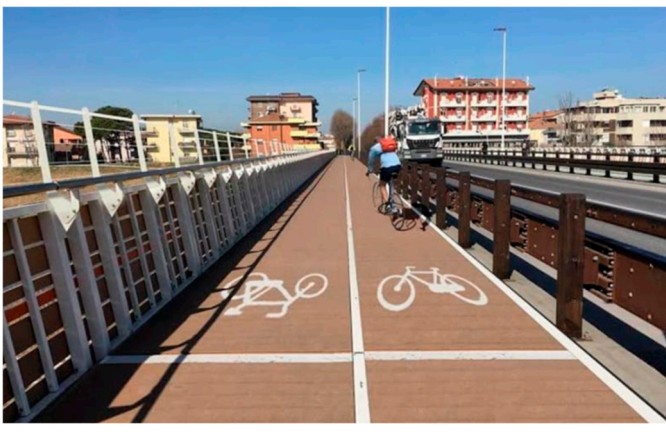

(**a**)        (**b**)

**Figure 2.** (**a**) Two-way cycle path on its own lane along Via Destra del Porto; (**b**) two-way cycle path on Ponte dei Mille [26].

Along the left bank of the canal, there is no cycle path parallel to the quay, but cyclists can ride without danger within zone 30 of the characteristic Borgo San Giuliano to reach the northern part of the city. Cyclists on the left side of the Canal Port can ride along this stretch on the carriageway intended for vehicular traffic.

It is possible to cross Rimini's Canal Port, at Ponte della Resistenza, Ponte dei Mille, and Ponte di Tiberio, or near the lighthouse, with a ferry boat, which is available to tourists and citizens both day and night in summer season, and also transports bicycles and pushchairs, connecting the two sides of Rimini's Canal Port.

Important aspects concerning soft mobility are covered by micro-mobility and sharing services. In the Rimini area, the vehicle stock at the disposal of tourists and Rimini residents comprises over 1300 e-scooters and 300 e-bikes. In addition, a call for tenders is being prepared for the experimentation of electric car-sharing services. These recently introduced shared vehicles support the transition towards sustainable mobility. Studies show that

the cycle path is widely used by e-scooters and e-bikes, with percentage values ranging from 60% to 90% [27]. Therefore, it is important that local authorities properly regulate the safe use of micro-mobility vehicles and develop proper policies for their integration into cities. The SUMP of Rimini already contains some indications regarding this issue. The Administration allows e-bikes and e-scooters a speed of 20 km/h on cycle paths and urban roads, and 6 km/h in pedestrian areas. On the basis of Law no. 15/2022, known as the "Milleproroghe Decree" [28], electric scooters are equated with velocipedes, and can therefore circulate in urban areas and on normal roads, respecting the same provisions governing the use of bicycles.

Rimini also offers a free-floating service with electric scooters. The convenience of this service lies in the lack of a specific pick-up or delivery point for the scooters, as they are distributed throughout the area enabled for their use. In addition to the free-floating pedal-assisted bikes, the summer of 2021 also saw the start of the electric scooter-sharing service, with around 50 vehicles made available as part of the call for tender for the installation of the columns.

The issue of carparks and parking areas assumes particular importance both in relation to the current mobility structure and in view of the new overall design of the mobility system, both public and private. The Municipality of Rimini has drawn up a Parking Plan, which, following an assessment of demand and supply in the various areas of the city, aims to increase supply. The strategic design of car parks assumes particular relevance in view of environmental sustainability. Indeed, intermodal cores are capable of expanding connections between inland and coastal areas in order to improve mobility and the quality of service offered, limiting pollutant emissions through a combination of soft mobility and reduced car use [29].

From all these analyses on transport systems, it emerged that the municipality of Rimini has already activated a reorganisation of infrastructure aimed at implementing soft mobility and micro-mobility, but there are still weaknesses around the Canal Port. In fact, while the right bank presents a good bicycle path, on the left side there is currently no dedicated path for cyclists. Furthermore, the quaysides are poorly connected to the upper pavement beside the carriageway. As a result, the area of the quaysides is poorly frequented because pedestrians walk along the walls without getting on the quays. These shortcomings also emerged from the surveys and the discussion with stakeholders and were the focal points on which the project was based.

## 3. Data Processing and Evaluations

The data collected from the surveys and preliminary analyses were classified as Strengths, Weaknesses, Opportunities, and Threats (SWOT). This SWOT matrix [30] was used as a data processing tool to gain an initial overview of all collected data.

In parallel, a questionnaire was submitted to the stakeholders involved in the decision-making process. The questions were related to the quality aspects of the Canal Port area from an environmental, economic, and social point of view. In this way, the area's critical issues and potentials were more clearly defined.

The combination of the data collected in the preliminary analysis phases, integrated with the feedback from the stakeholders, allowed the identification of a set of indicators that, appropriately calculated, provided an insight into the urban development situation of the Canal Port [31]. These indicators, identified and assessed by Casamassima [32], constitute the basis for the subsequent project phase.

The project actions for urban regeneration identified in the data processing phase were numerous. In order to prioritise them, a Benefits, Opportunities, Costs and Risks (BOCR) model was used, taking into consideration the different aspects of the decision-making process: economic, environmental, social, infrastructural and urban aspects [33]. The following subsections describe the data processing steps.

### 3.1. SWOT Analysis

SWOT (Strengths, Weaknesses, Opportunities, and Threats) analysis is a frequently used tool for developing strategic planning. Particularly appreciated in urban planning, SWOT analysis assesses internal and external factors, as well as the current and future potential of the project area. Even if it is extremely useful for a first phase of processing and interpretation of the data collected from the analysis of the state of the art, it does not provide information on the degree of priority of an intervention over the others [34]. The data collected in the first analysis phase were sorted into a SWOT matrix. From the outcomes that emerged as opportunities, the project alternatives were derived as inputs into the BOCR model to prioritise project actions. Moreover, the meta-design analysis was deepened through the use of indicators that allowed the identification of a priority action degree to be taken for the redevelopment of the Canal Port area through a BOCR analysis.

### 3.2. Stakeholders' Involvement

As demonstrated by the Inter-Connect Project, integrated mobility between different modes of transport—road, rail, and sea, but also pedestrian and cycle—allows a considerable reduction in pollutants from vehicle traffic, moving towards sustainable mobility [35]. This is achieved by considering various sustainability factors, such as social, economic and environmental aspects. These were also investigated in this data processing phase through a dialogue with stakeholders, by analysing aspects such as: urban quality, safety and comfort of transfers, economic well-being of productive and commercial activities, and environmental quality [36].

During the data collection phases, a survey was carried out among the stakeholders involved. A questionnaire was distributed to them in order to identify the most critical aspects and to quantify the qualitative aspects essential for assessing urban quality. A multiple-choice questionnaire was sent online, not only to public or private bodies, but also to all the actors who make daily use of the services of the Canal Port, such as the Authority Italian Naval League, the Nautical Club of Rimini, the Sea Workers' Cooperative, and Rimini Sailing Club. The questionnaire was divided into two sections: Section 1 about infrastructure and transport systems and Section 2 about public space. In the end, two open questions asked about the phenomena of urban and social degradation and about the main shortcomings and/or criticalities of the area under consideration.

The following conclusions can be drawn from the results of the questionnaire:

- Trips with private motorized vehicles are safe and the infrastructures are on average perceived as satisfactory, but there are considerable inconveniences due to traffic.
- Traveling by public transport is considered safe, but shortcomings in intramodality have emerged.
- The cycle/pedestrian paths are generally well-lit and signposted and are perceived as being generally safe, but scarce shade has been reported, and in general, they are considered to be not completely adequate to the needs of cyclists and pedestrians.
- The parking lots were evaluated very negatively in almost every aspect investigated. Particular issues that emerged the scarcity of available lots and the amount of time needed to find a parking lot.
- The public spaces generally meet the expectations of the stakeholders in the aspects of lighting and safety, but some unsatisfactory elements remain, such as the scarce presence of green and urban furniture, the ineffective integration of the Canal Port area with the urban landscape of Rimini, poor cleaning and maintenance, and the presence of architectural barriers.
- The water quality of the Canal is considered very low due to dirt and the lack of water recirculation.

In addition to the questionnaire, also useful for statistical purposes in the elaboration of the set of indicators for urban regeneration, the dialogue with stakeholders was constant and continuous throughout the entire preliminary project phase. Regular meetings ensured

that the project proposals were adapted to the needs of the stakeholders and end-users to achieve more effective targeted solutions [37].

### 3.3. Multi-Criteria Analysis

In order to assess urban quality, support the design choices, and the monitoring phase of the proposed interventions, it was decided to identify a set of indicators: this solution allows for a qualitative–quantitative assessment of the various aspects that contribute to pursuing the objective of a sustainable city. The indicators identified for the study area were classified according to five categories in order to analyse not only the aspects included in the Venn diagram, but also the infrastructural aspects related to transport and those of urban morphology related to the context (Figure 3). The decision to integrate these other two categories with those already present in the conventional Venn diagram was made because the urban and transport system has a decisive influence on urban well-being and, consequently, on the sustainability of the project intervention.

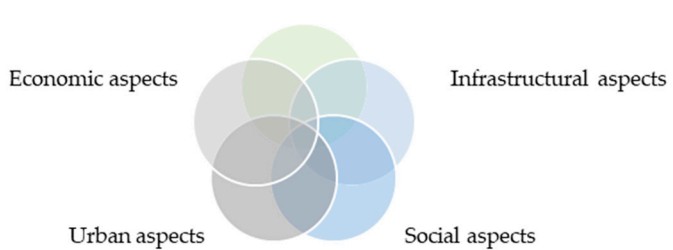

**Figure 3.** Indicators categories (© 2022, R. Corticelli—CIRI Building and Construction).

The evaluation of alternative urban transformation scenarios represents a complex decision-making problem that is frequently analysed by means of Multi-Criteria Analysis (MCA). In this framework, the Analytic Network Process (ANP) methodology plays a prominent role [38]. Developed by the American scholar Thomas L. Saaty [39], it represents the generalisation of the simpler Analytic Hierarchy Process (AHP) methodology [40] to problems of a more complex nature that encompass various degrees of interaction between the analysed elements. In AHP, the decision problem is schematised as a network of elements organised into groups and correlated by various influence relationships. The network structure allows the assessment of interdependence relationships both within each group of elements and between the various groups of elements [41].

A common case of a complex network model with control hierarchies giving rise to sub-networks is the Benefits, Opportunities, Costs, Risks (BOCR) model [42], which, similarly to the Strengths, Weaknesses, Opportunities, Threats (SWOT) matrix, refers to two time dimensions:

1. Benefits and Costs are measured at present time.
2. Opportunities and Risks are estimated on the basis of expectations of the impacts of the intervention in the long term.

In this model, the complexity of the problem is decomposed into four sub-networks: Benefits, Opportunities, Costs, and Risks. Each of these four sub-networks contains five clusters of environmental, economic, infrastructural, urban, and social aspects. Each sub-network produces a ranking of alternatives that will then be correlated with those of the other sub-networks to obtain an overall result that provides an overall ranking of choice options [43].

In the case of the Rimini Canal Port, the intention was to assess the priority of intervention among the redevelopment actions identified by the preliminary analyses. The possible alternatives among which the design options were selected are:

- Option 0: maintaining the current configuration of the Canal Port area. i.e., the no-intervention option;
- Option 1: creation of urban spaces of better quality;
- Option 2: implementation of the "Traghetto Vittoria" ferry service;

- Option 3: construction of a new interchange car park;
- Option 4: reconnection of cycle and pedestrian paths;
- Option 5: redevelopment and raising of docks;
- Option 6: construction of the new Fish Market;
- Option 7: construction of new tourist connections;
- Option 8: redevelopment of the slipway.

From the outcomes of the BOCR analysis, the two intervention alternatives that emerged as priorities are **the reconnection of cycle and pedestrian paths** (17.28%), and **the redevelopment and raising of docks** (16.83%). Since these two options were almost on the same ranking level and are closely interconnected, a discussion with the Municipality of Rimini revealed a willingness to solve these two problems within the same urban regeneration project.

## 4. Urban Regeneration Project

Once the priority actions were identified—the reconnection of cycle and pedestrian paths, and the redevelopment and raising of docks—the design phase of urban regeneration started. The raising of docks was needed to avoid flooding caused by a mistake in the last requalification project dating back to 1977. It was established that it was necessary to solve this problem and to reconnect the cycle/pedestrian network with the urban transport infrastructure, the preliminary project led to the development of a master plan and the identification of social activities to be located in the new-created public spaces.

### 4.1. Identification of the Raising Height

The most critical issue encountered in the project area is the frequent flooding of the docks due to tides and adverse marine weather conditions. In order to solve this problem, which is due to a design mistake dating back to 1977, the only option is raising the height of the quaysides, taking into consideration both the sustainable mobility aspects of bicycle and pedestrian paths and the identification of new spaces and activities for the benefit of Rimini's citizens and tourists.

For the identification of the quayside elevation height, the report produced by CESI on commission by Alpina Acque s.r.l. [44] on the occasion of the Marecchia Park hydraulic project was consulted. It includes the hydraulic model of the Canal Port from the Tiberius Bridge to the sea. It contains a collection of extraordinary (uncommon) simulation scenarios that reproduce the combination of various hydraulic and meteorological phenomena that may affect the flood flows drained by the Marecchia River. The Marecchia river flows in the Tiberio basin within a speed range from 200 $m^3$/s to 300 $m^3$/s, and this is considered in combination with various sea level rises at the port mouth, but in these combined scenarios, the return times, i.e., the frequency of the two events combination, have not been considered.

Observing the two variables individually, the hydraulic conditions capable of producing the flow rates simulated in the extraordinary scenarios of this study (200–300 $m^3$/s) into the historical riverbed (and subsequently into the Tiberio basin) are characterised by return times greater than 200 years.

With regard to the return times of sea-level rise, however, it appears that a tidal rise of 1.30 m occurs annually, one of 1.70 m occurs every 10 years, while a longer return time of 100 years predicts a maximum tidal rise of 2.10 m. The combination of the two scenarios will give a certain level of risk resulting from the probabilistic combination of the simultaneous occurrence of both events.

In addition to these two parameters, it is also necessary to take into account the subsidence phenomenon, which, according to a consultation with Massimo Paganelli (Municipality of Rimini), is about 0.5–1 cm per year (in the last 50 years it has been about 50 cm). Moreover, it is necessary to consider the current height of the quays also in relation to the height above sea level (a.s.l.) of the context they refer to. From the topographical survey carried out by Geo-Graphic (Geom. A. Bertozzi and M. Stabellini) commissioned by the ITL Foundation, it can be noticed that in correspondence with the railway bridge

the height of the docks is currently at 0.32 m a.s.l., while the height of the road is between 0.46–0.61 m a.s.l., as shown in Figure 4.

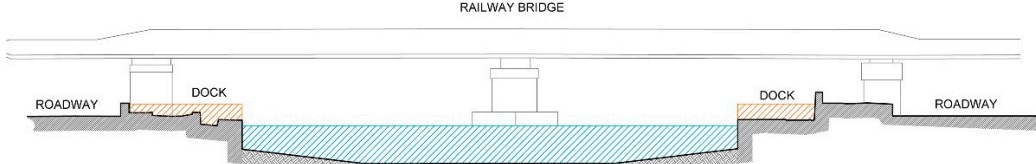

**Figure 4.** Dock elevation near the railway bridge (© 2022, R. Corticelli—CIRI Building and Construction).

This section is the most critical and frequently flooded. As it is impossible to change the height of the road surface, since it is necessary to guarantee the passage of vehicles under the bridge, it is proposed to raise the height of the quaysides to the height of the cycle path, which, in the section near the railway bridge, is at 1.50 m a.s.l.

From the consultations with the Emilia–Romagna Region's technicians and the study of the material provided by the Municipality of Rimini, it was deducted that it was necessary to raise the docks to a height between 1.30 and 1.70 m a.s.l. Such a height represents the optimal compromise for the safety of quays in the long term, since it also takes into account the subsidence phenomenon, while respecting the urban context in which the intervention takes place. Specific solutions need to be assessed to allow access to mooring (e.g., by means of floating platforms). The docks downstream from Ponte della Resistenza are at an elevation of 1.24 m a.s.l. and have been recently renovated; therefore, they have not been considered in the project.

To conclude, it is reported that during a meeting with the Emilia–Romagna Region's technicians it was identified that there is a need to conduct an updated hydraulic study to validate the correctness of the proposed elevation. In fact, raising the docks to 1.50 m a.s.l. would not definitively solve the problem of flooding, but would considerably lengthen the return time of the occurrence of such events.

While waiting for the updated study to be carried out, the project proposal illustrated in this contribution is based on raising the docks to 1.50 m a.s.l. in the sections from Ponte della Resistenza to Ponte di Tiberio.

*4.2. Study of Critical Sections*

As mentioned, downstream of Ponte della Resistenza the quays have already been recently renewed by the Municipality of Rimini. In that stretch, the docks are never lower than 1.20 m a.s.l. and are not flooded as frequently as in the stretch upstream from Ponte della Resistenza Bridge. For this reason, the design proposal focused on the quays between Tiberius Bridge and Ponte della Resistenza Bridge, highlighted in Figure 5.

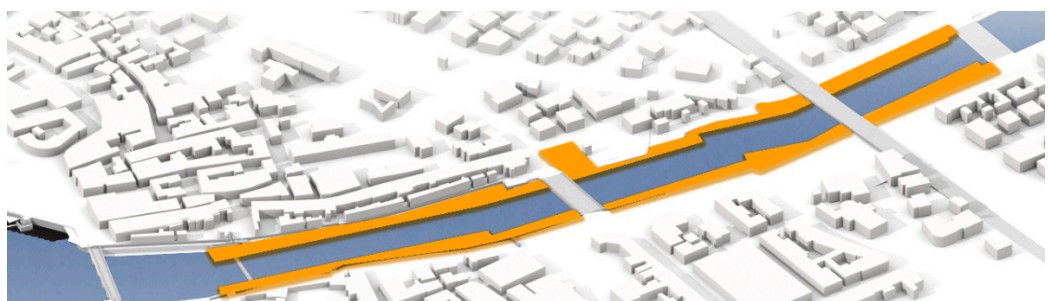

**Figure 5.** Stretch of quays to be raised (© 2022, R. Corticelli—CIRI Building and Construction).

Between Tiberius Bridge and Ponte dei Mille Bridge there are currently mooring bollards, which, although currently unused, are elements of historical–documental value, dating back to the dock renewal work carried out in 1977, while between Ponte dei Mille and Ponte della Resistenza, boats belonging to the local association Amici del Mare (Friends

of the Sea) are currently moored. In order to regularise these moorings, which are currently unauthorised, and to allow access to the boats after the renovation work to raise the docks, the following design solutions are proposed:

- Raised dock at an elevation of 1.50 m above sea level; the dock would be raised entirely and there is no access for boats.
- Raised dock at an elevation of 1.00 m above sea level: the strip of quay closest to the edge of the canal would remain at an intermediate level between the current level and 1.50 m to allow access to boats. It is possible that this strip will be submerged at times during the year, but less frequently than at present.
- Floating platform: in those sections where it is not possible to dedicate a strip of the quay to mooring access, it is proposed to install floating platforms made of medium-density polyethylene anchored to the quay by means of steel brackets. These platforms move vertically along the steel guides following the free surface of the water.

To verify the effectiveness of the design solutions, six significant sections affecting the stretch of the Canal Port between the Bridge of Tiberius and the Ponte della Resistenza were analysed (Figure 6).

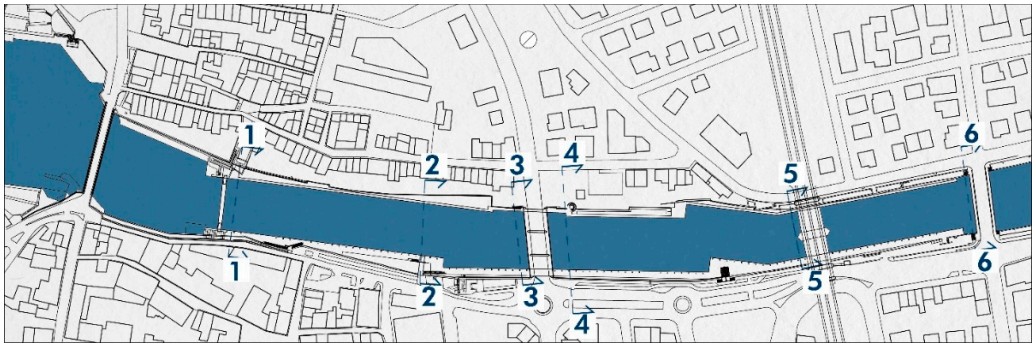

**Figure 6.** Six most significant sections (© 2022, R. Corticelli—CIRI Building and Construction).

All sections were made looking downstream towards the sea outlet.

Section 1-1 (Figure 7) immediately downstream of the moving footbridge includes the panoramic terrace and the cycle/pedestrian walkway on Via Bastioni Settentrionali. Here, the quay would be raised from an elevation of 0.33–0.39 m a.s.l. to an elevation of 1.50 m a.s.l. The staircase that currently descends on the quay from Via Bastioni Settentrionali towards the Tiberius Bridge would be closed, allowing the widening of the cycle-pedestrian walkway, and leaving the sail designed by architect Vittoriano Viganò as evidence of the architectural brutalism of 1977. The only access ramp to the quay from the right side of the canal would be the other ramp that descends towards Ponte dei Mille.

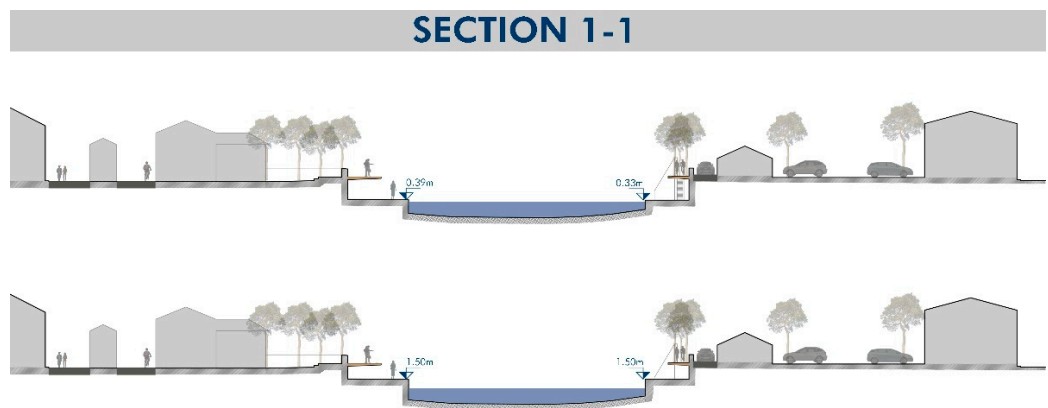

**Figure 7.** Section 1-1: before and after dock elevation (© 2022, R. Corticelli—CIRI Building and Construction).

Section 2-2 (Figure 8) at Vittoriano Viganò's panoramic terrace shows that a raising of the docks would prevent the passage under the existing terrace. Considering the fact that access to this space is currently denied and that it is therefore not used, it is proposed to demolish it and replace it with a light steel and wood ramp similar to the cycle/pedestrian walkway in Via Bastioni Settentrionali. In this way, the space created on the quay can be dedicated to green flowerbeds and small temporary shops.

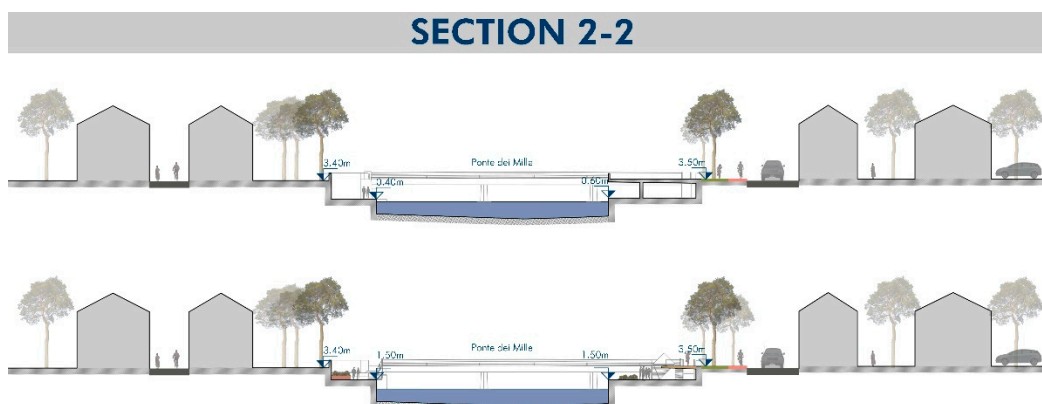

**Figure 8.** Section 2-2: before and after dock elevation (© 2022, R. Corticelli—CIRI Building and Construction).

Section 3-3 (Figure 9), immediately upstream Ponte dei Mille, shows the proposal to prevent passage under the bridge and to build ramps to permit passage over it. The design choice is motivated by the fact that raising the quaysides would compromise the height necessary to allow passage under the bridge. To avoid the creation of a subway that would frequently flood, interrupting the continuity of the paths on the quays, and which would constitute a hidden point potentially subject to social decay, it is preferable to build ramps going up on Ponte dei Mille. Such ramps would also provide an additional link between the upper streets beside the canal and the quays, and can, therefore, help to increase the number of pedestrians on the quays. On the right side of the canal, Porta Galliana has recently been restored, and a connection to the Ponte dei Mille may also be advantageous to entice visitors to continue their walk on the quays.

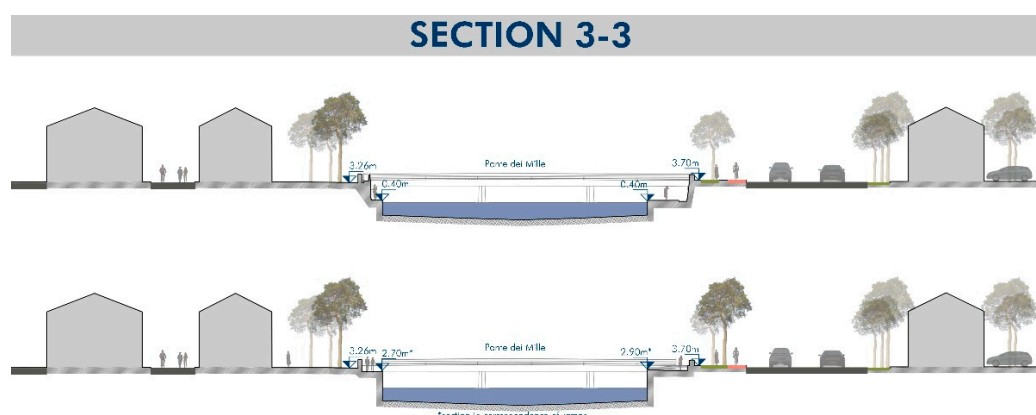

**Figure 9.** Section 3-3: before and after dock elevation (© 2022, R. Corticelli—CIRI Building and Construction).

Section 4-4 (Figure 10), downstream from Ponte dei Mille, shows, on the left side of the canal, the new conformation of the Don Luigi Sturzo Gardens. It is proposed to demolish the wall housing the spiral staircase, which is currently a drug dealing site, and to create a square on three levels sloping down towards the canal. This multi-levelled space would create several striking views and channel the flow of pedestrians towards the quays. The square would be integrated with more trees in addition to those already present and with play areas for children, to make it a focal point for the Borgo San Giuliano community.

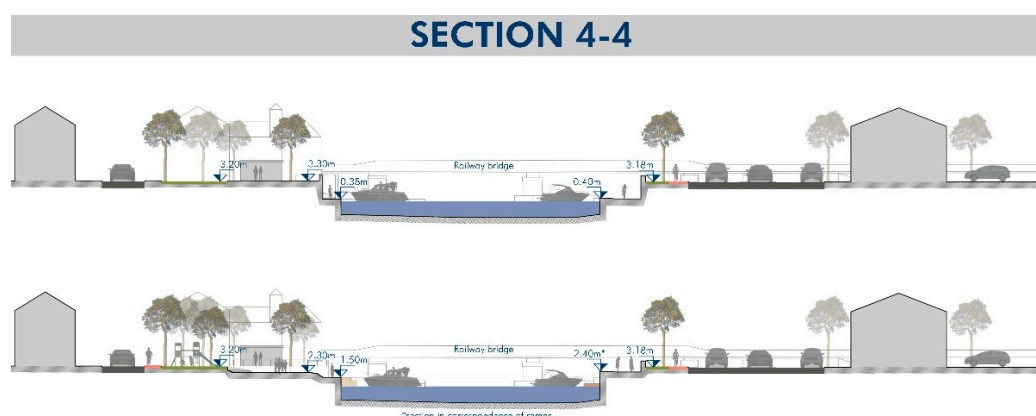

**Figure 10.** Section 4-4: before and after dock elevation (© 2022, R. Corticelli—CIRI Building and Construction).

Section 5-5 (Figure 11), upstream of the railway bridge, on the left side of the canal, shows the proposed method of raising the quay to an elevation of 1.50 m a.s.l., eliminating the current terracing and creating a single space. An aromatic herb garden and some trees would be placed here, among which it would be possible to walk. On the right side, by contrast, it is proposed that a portion of the city walls not under constraints should be demolished, bringing the height of the quays to the current level of the cycle path pavement on Via Destra del Porto. This solution allows for a single space that is more usable by pedestrians and cyclists.

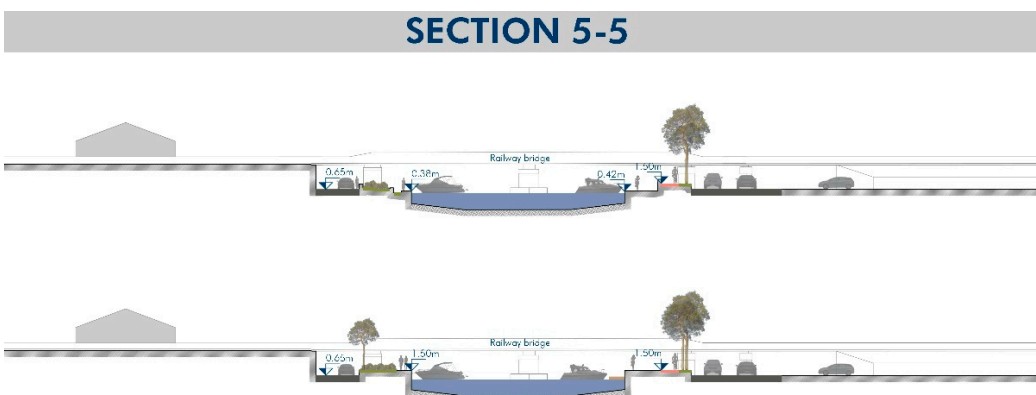

**Figure 11.** Section 5-5: before and after dock elevation (© 2022, R. Corticelli—CIRI Building and Construction).

Section 6-6 (Figure 12), upstream of Ponte della Resistenza, shows the proposal to raise the height of the quays to the level of the roadway on both sides of the canal. Since such an elevation would make boat access impossible, or extremely uncomfortable, and since this section of the canal is too narrow to accommodate floating platforms, in this case, it is proposed to set a strip of quay with an average width of 1–1.20 m at an elevation height of 1.00 m a.s.l. to allow mooring and boat access.

### 4.3. Design of Dock Access Ramps

After the examination of the most significant sections, the focal aspect of the urban regeneration project was the design of the ramps to connect the docks to the urban cycle/pedestrian network. Each access point was carefully analysed, trying to find solutions to overcome architectural barriers and to improve the and accessibility of the spaces (Figure 13).

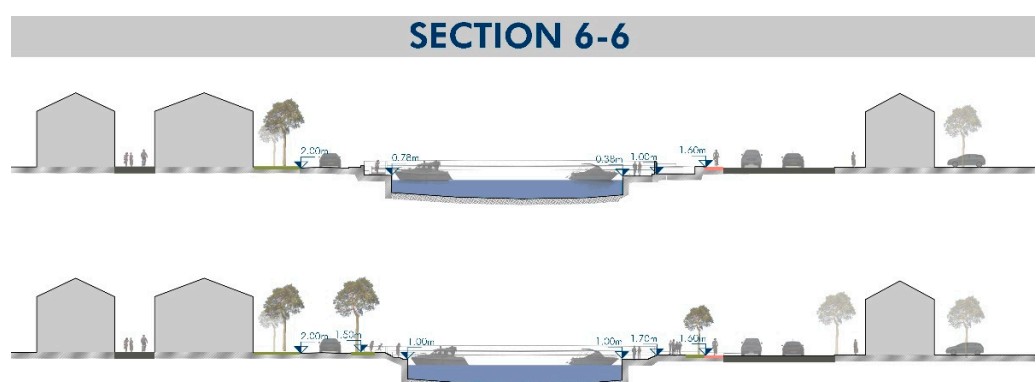

**Figure 12.** Section 6-6: before and after dock elevation (© 2022, R. Corticelli—CIRI Building and Construction).

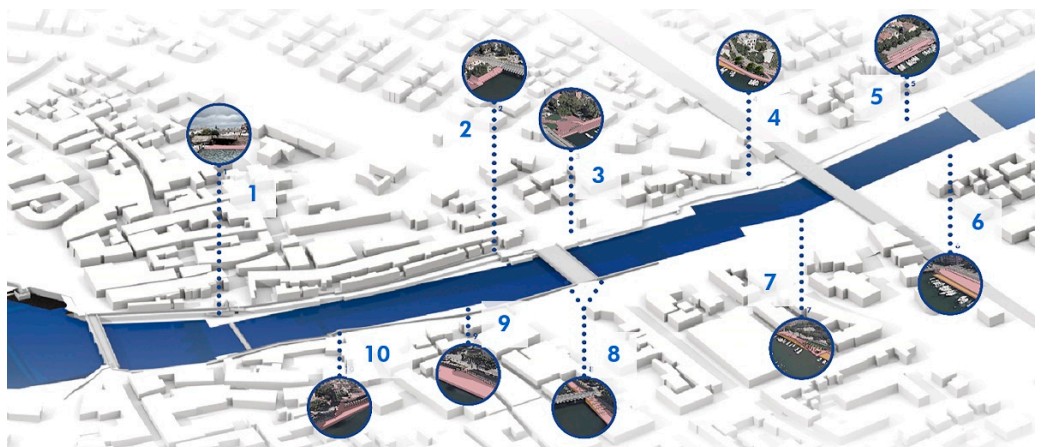

**Figure 13.** Access points to the docks (© 2022, R. Corticelli—CIRI Building and Construction).

The ten access points identified to improve the accessibility and feasibility of the docks are reported and described below:

- Access 1: ramp at the mobile pedestrian footbridge nearby the panoramic terrace;
- Access 2: at Ponte dei Mille besides the Chinese Restaurant "Fiore di Loto";
- Access 3: through the Giardini Don Luigi Sturzo;
- Access 4: connection between Via Madonna della Scala and the docks at the Railway Bridge;
- Access 5: at Ponte della Resistenza, between Via E. Coletti and Via Sinistra del Porto;
- Access 6: at Ponte della Resistenza, between Via E. Coletti and Via Destra del Porto;
- Access 7: connection between Via G. Savonarola and the quay at the Railway Bridge;
- Access 8: new access to be realised to connect the docks to the Ponte dei Mille;
- Access 9: through the panoramic terrace near Porta Galliana;
- Access 10: in Via Bastioni Settentrionali, at the entrance to the cycle/pedestrian path.

### 4.3.1. Access 1

The footbridge and the panoramic terrace in Piazzetta Pirinela are part of the "Tiberio compartment 4 Canale" project, a plan implemented by the Municipality of Rimini to redevelop the Tiberius Bridge.

The ramp at the mobile pedestrian footbridge on the side of the panoramic terrace is the most direct connection to access the docks from Borgo San Giuliano. It was built recently and complies with the regulations governing accessibility and the removal of architectural barriers. With the raising of the quays to a height of 1.50 m a.s.l., it will be necessary to adjust their length and slope to allow landing on the raised quay (Figure 14).

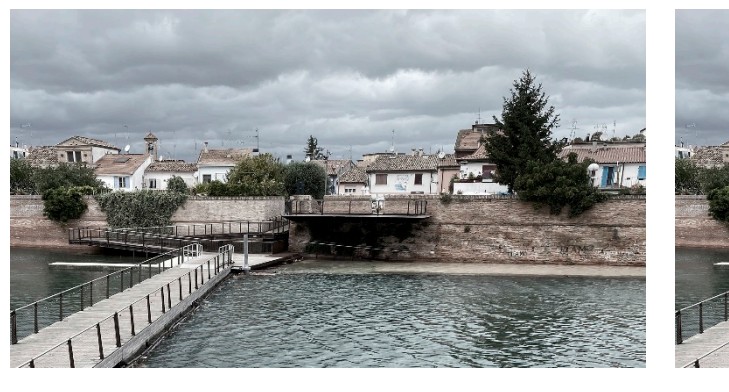 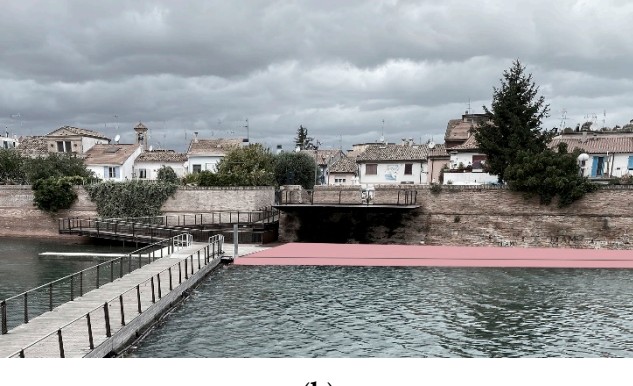

(**a**)　　　　　　　　　　　　　　　　　　　　　　　　(**b**)

**Figure 14.** (**a**) Before, and (**b**) after dock elevation at Access 1. The pink-highlighted area shows the portion of quay subject to superelevation (© 2022, R. Corticelli—CIRI Building and Construction).

### 4.3.2. Access 2

The staircase that currently provides access to the quay from Ponte dei Mille constitutes an architectural barrier and is scarcely visible. The hidden position in the pertinence area of the Chinese Restaurant "Fiore di Loto" and the poor accessibility hinders its use by the public. Since the raising of the quay would prevent passage under Ponte dei Mille, it is proposed to demolish the currently existing staircase and replace it with a ramp. It would have a maximum slope of 8% and land directly on Ponte dei Mille. This ramp could be partially stepped and accommodate seats that would allow users to pause and admire the view of the last stretch of the Canal Port before the Bridge of Tiberius. (Figure 15).

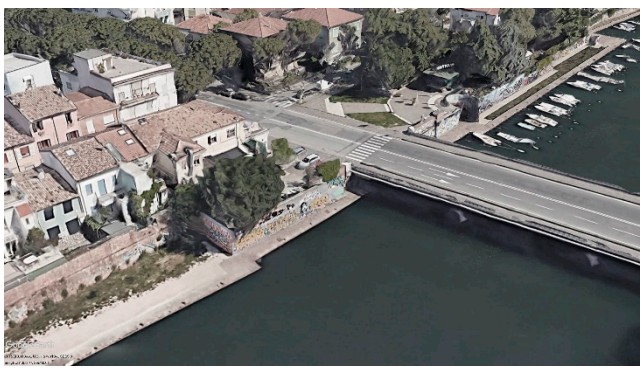 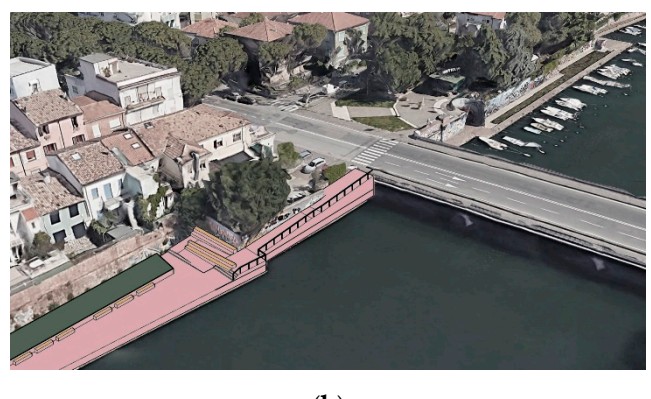

(**a**)　　　　　　　　　　　　　　　　　　　　　　　　(**b**)

**Figure 15.** (**a**) Before, and (**b**) after dock elevation at Access 2. The pink-highlighted area shows the portion of quay subject to superelevation (© 2022, R. Corticelli—CIRI Building and Construction).

### 4.3.3. Access 3

The spiral staircase that provides access to the quayside from Don Luigi Sturzo Gardens is a drug dealing site that shows serious signs of degradation, and due to this lack of attractiveness is barely used.

To eliminate the blind spot below the staircase and make the area more open, solving at the same time the problem of drug dealing and making the access passage to the quays more accessible and visible, it is proposed to open a portion of the retaining wall of the canal and to create a square connecting the height of the raised quays with that of the current roadway. An operation of this kind would, in fact, allow the pedestrian path to be naturally channelled towards the quays, obtaining a greater use of these spaces and a better urban quality (Figure 16).

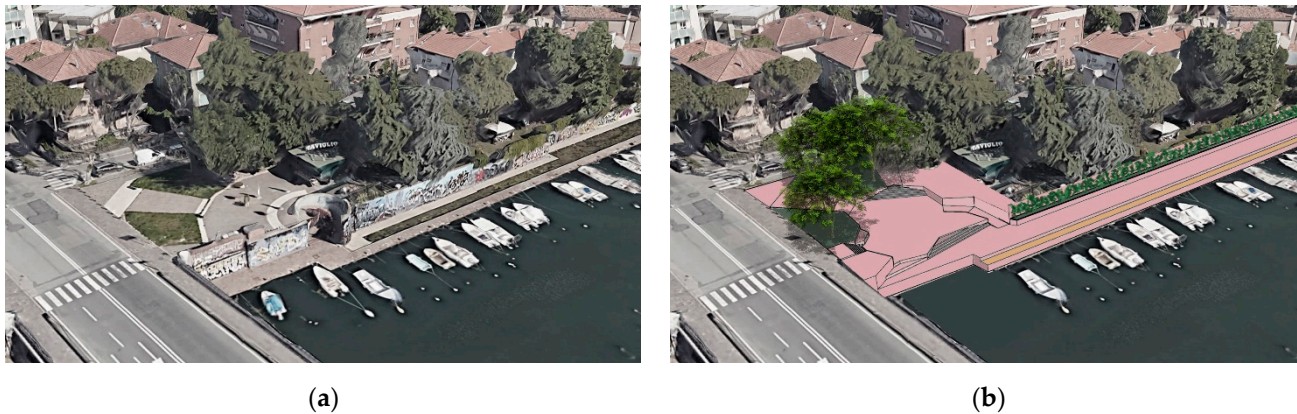

(**a**)            (**b**)

**Figure 16.** (**a**) Before, and (**b**) after dock elevation at Access 3. The pink-highlighted area shows the portion of quay subject to superelevation (© 2022, R. Corticelli—CIRI Building and Construction).

The quay in this section would be raised, covering the drainage channel that is currently saturated with gravel. In order to allow and regulate the access of the small boats of the local association Amici del Mare (Friends of the Sea), the strip closest to the boats would not be raised to a height of 1.50 m a.s.l., but left at a height of 1.00 m a.s.l. and covered with wood plastic composite (WPC), a product composed of bamboo and additives that maintains the aesthetic appearance of wood but is more durable and requires less maintenance.

### 4.3.4. Access 4

Via Madonna della Scala is connected to the quay at the Railway Bridge by a small, barely visible, and practically unused staircase. A series of terraces with concrete walls form an architectural barrier that discourages access to the quay.

Since the dock raising would bring it to the level of the current roadway, and since the surveys revealed the absence of a safe cycle/pedestrian passage in this section, it is proposed to build a cycle path along Via Madonna della Scala and a redesign of the flowerbeds to allow access to the quay in a more comfortable way. The planting of three small trees could also encourage greater use of this urban green space. In the corner on the quay behind the Church of Madonna della Scala, the installation of a WPC floating platform is proposed. This would allow easy access to the moorings, even when the water level in the canal is low without reducing the pedestrian passage on the quays, which is particularly narrow at this point. Floating platforms can be attached to existing docks via steel guides. This allows them to move freely along the vertical direction, making it unnecessary to remove them in the event of the Marecchia River draining into the Canal Port (Figure 17).

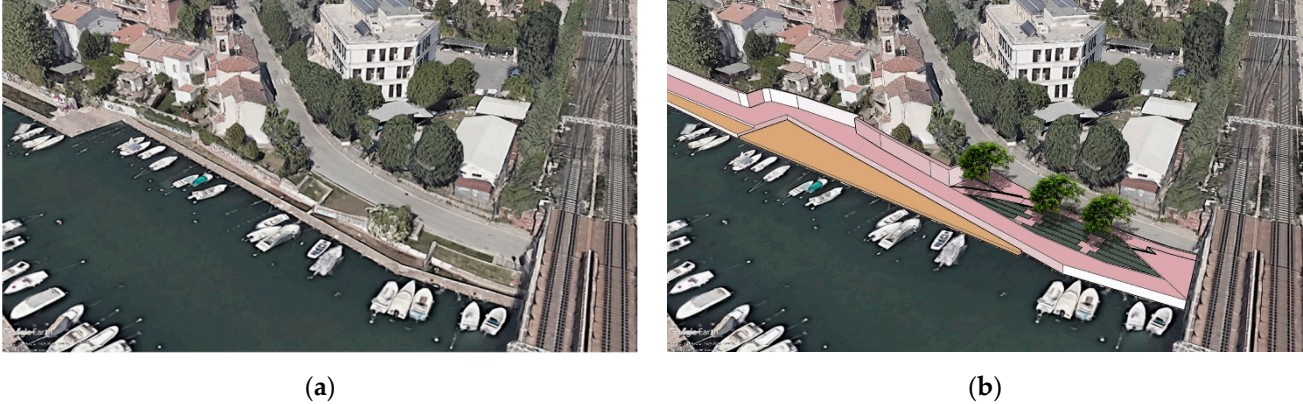

(**a**)            (**b**)

**Figure 17.** (**a**) Before, and (**b**) after dock elevation at Access 4. The pink-highlighted area shows the portion of quay subject to superelevation (© 2022, R. Corticelli—CIRI Building and Construction).

### 4.3.5. Access 5

Between Via Sinistra del Porto and Ponte della Resistenza, there is currently a difference in height between the pavement at the roadside and the level of the quay. The section towards the railway bridge has flowerbeds with maritime pines.

Also in this section, raising the quay would bring it to the same level as the road level. Therefore, it is proposed to create a single pedestrian space, including both the quay and the pavement along Via Sinistra del Porto. The space would accommodate public spaces, with seating surrounded by greenery, and the flowerbeds already present would be expanded without removing the pine trees. The activities that have been selected to be placed on the docks are based on the assumption that physical mobility is fundamental, not only from the perspective of transport sustainability, but also from the point of view of physical activity. For this purpose, this stretch of quayside hosts a calisthenics park that, as with the other proposed soft mobility solutions, provides exercise opportunities that have a multitude of positive impacts on human health [45].

As in the section of the quay in correspondence with the Don Luigi Sturzo Gardens, a strip should be left at an altitude of 1.00 m a.s.l. to allow access to moored boats when the water level in the canal is low. This strip can also be paved with WPC, again to match the appearance of the floating platforms and better distinguish the mooring spaces (Figure 18).

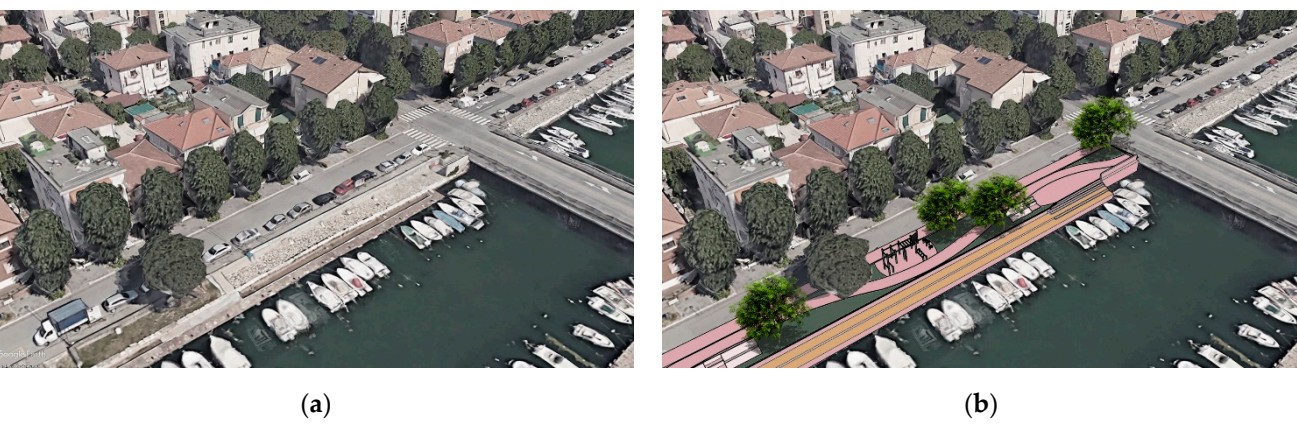

(**a**)                                                                                    (**b**)

**Figure 18.** (**a**) Before, and (**b**) after dock elevation at Access 5. The pink-highlighted area shows the portion of quay subject to superelevation (© 2022, R. Corticelli—CIRI Building and Construction).

### 4.3.6. Access 6

Access to the quay between Via Destra del Porto and Ponte della Resistenza is currently provided by a flight of steps leading to a wide stone pavement. This is connected to the porphyry quayside by a flight of steps in Istrian stone, which is, however, currently interrupted by the drainage ditch. The current conformation of this section of the quay, together with the absence of functions within it, makes it completely unused, even though it is a large space potentially capable of hosting a multitude of functions.

Raising the dock to 1.50 m a.s.l. allows the elimination of the flight of stairs between the roadway and the quay, leaving only a ramp necessary to connect the 1.00 m high strip mooring access to the elevated dock. The raised quay provides ample free space that can accommodate commercial activities such as bars and kiosks that would help to populate the quaysides with people.

The Istrian stone staircase, even if not listed in the PSC, is an element of architectural value, since it is an intervention made on the ancient walls. For this reason, it is proposed to dismantle it, clean it up and relocate it as an access ramp to the raised quay that remains at an altitude of 1.00 m a.s.l. to allow access to boats. Furthermore, since this is a large area to be used as a public space, it is proposed the installation of two flowerbeds with bushes to provide shade without obstructing the view of the canal (Figure 19).

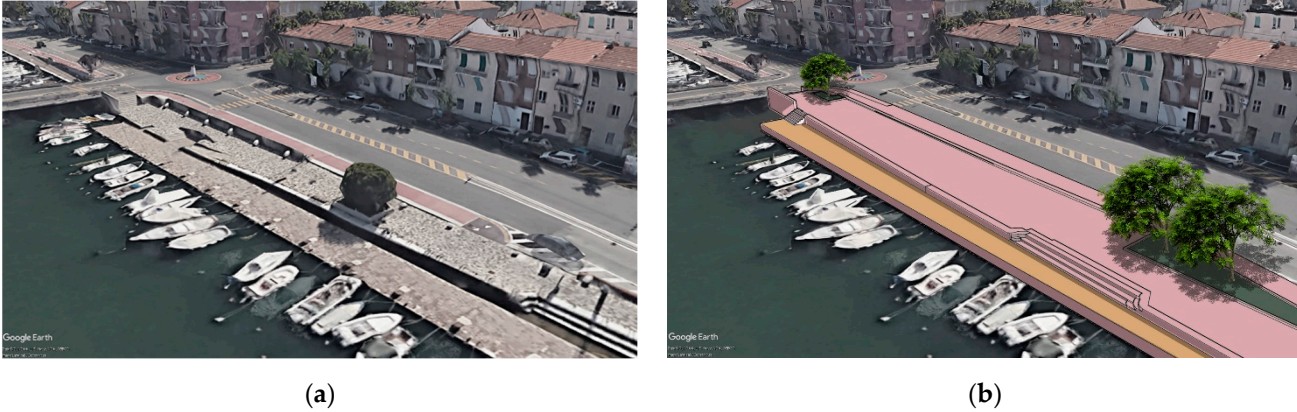

| (**a**) | (**b**) |

**Figure 19.** (**a**) Before, and (**b**) after dock elevation at Access 6. The pink-highlighted area shows the portion of quay subject to superelevation (© 2022, R. Corticelli—CIRI Building and Construction).

### 4.3.7. Access 7

A staircase designed by architect Vittoriano Viganò and built in 1977 allows access to the dock. It constitutes a considerable architectural barrier: the shape of the existing ramp makes it unattractive to pedestrians, who usually avoid it by staying on the pavement above the walls and not descending onto the quay.

Since this ramp is currently completely unused and in a state of decay, and considering that an elevation of the quay would in any case imply work on it, it is proposed to completely eliminate this ramp in order to make better use of the free space that would be created on the quay for the benefit of the community.

As shown in Annex C2—Draft board C.1.1.4 "Carta della tutela monumentale" (Monumental conservation map) of Rimini's PSC [46], the monumental constraints protecting the city walls do not affect the tract of the wall near Railway Bridge. Therefore, it is proposed to demolish a stretch of the wall that is not under constraint, and to bring the quay to the same level as the existing cycle path. In this way, the physical and visual unity of the spaces would invite users to make greater use of the quayside spaces (Figure 20).

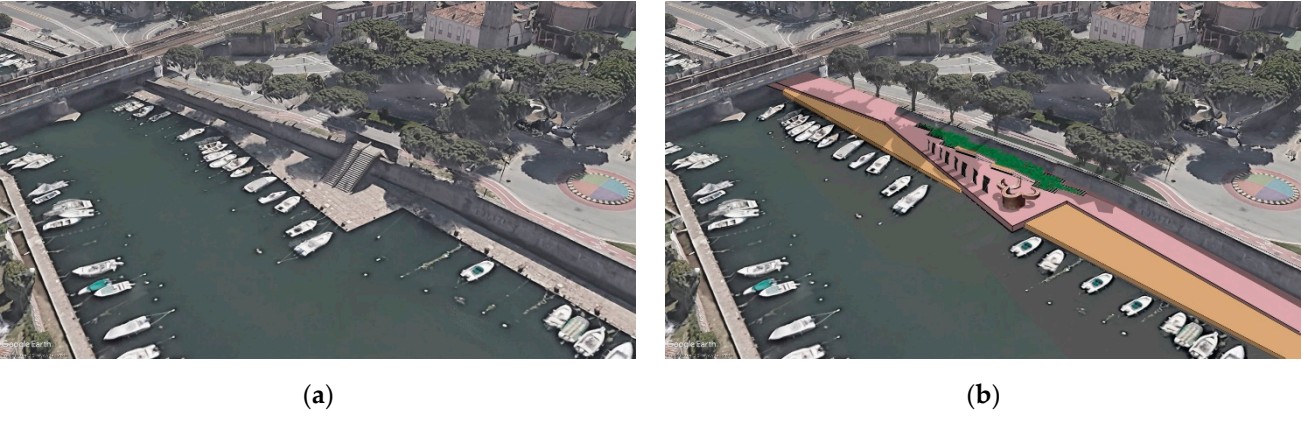

| (**a**) | (**b**) |

**Figure 20.** (**a**) Before, and (**b**) after dock elevation at Access 7. The pink-highlighted area shows the portion of quay subject to superelevation (© 2022, R. Corticelli—CIRI Building and Construction).

### 4.3.8. Access 8

There are currently no access ramps connecting Ponte dei Mille to the quay on the city-centre side. With the raising of the quaysides to 1.50 m a.s.l., it would be impossible to maintain a useful height for passing under the bridge. This would therefore require the construction of a ramp to pass over it. From the survey of the quay, which in this section is about 6.00 m wide, it would be possible to build an alternating sloping ramp with steps that would meet the requirements of the regulations governing accessibility and the removal

of architectural barriers [47]. Such a ramp in itself constitutes an encouragement to access. In fact, its conformation leaves the user the freedom to walk along it in the preferred way, and, if necessary, to stop there using the steps as a seat. This freedom and adaptability in the use of space invites users to take possession of urban spaces [48].

The ramp must be suitably protected, on the side facing the canal, by a parapet at least 1.00 m high that cannot be crossed by a 10 cm diameter sphere. If there is a non-solid parapet on the side of the ramp, the ramp must have a kerb at least 10 cm high. The only caution, in this case, is that the raising of the quay and the ramp to the bridge cannot be built in adherence to the historic wall, since this is a listed heritage asset. Therefore, a solution should be found during the design phase that allows the construction of these elements while maintaining a suitable detachment from the listed property. The simplest solution is to identify a suitable separation distance between the ramp and the historic wall, and to cover the cavity with a rigid Polyvinyl chloride (PVC) or pre-painted aluminium-covering profile 20/10 thick. The location of rainwater drainage channels should also be evaluated (Figure 21).

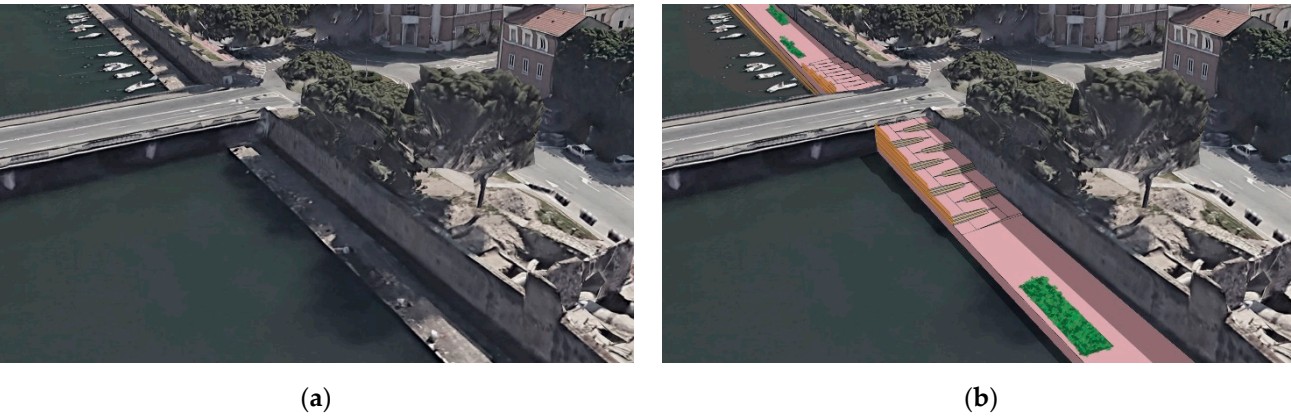

(**a**)                                                                                                   (**b**)

**Figure 21.** (**a**) Before, and (**b**) after dock elevation at Access 8. The pink-highlighted area shows the portion of quay subject to superelevation (© 2022, R. Corticelli—CIRI Building and Construction).

4.3.9. Access 9

The access at Porta Galliana was also built in 1977, as evidenced by the Vittoriano Viganò design. The staircase is currently inaccessible for security reasons. Apart from the serious state of degradation, the access point is not suitable to meet the requirements of the standard for overcoming architectural barriers, and should be revised to adapt it to the new height of the raised quaysides.

For this reason, in line with that proposed for access 7, it is proposed to demolish the staircase and terrace. They are not subject to constraints and their demolition allows, also in this case, to dispose of the space that is being opened on the quay for the benefit of the community. The elimination of the panoramic terrace is not a disadvantage, since the view of the Tiberius Bridge is almost identical to that from the raised quay.

Instead of the panoramic terrace in Viganò, it is proposed to build a light steel staircase with wooden flooring similar to the cycle/pedestrian walkway along Via Bastioni Settentrionali, in order to promote the perception of a unified and integrated urban space that is at the same time qualitatively valuable. The new access point has on one side a flight of stairs with a resting landing halfway down the ramp, while on the opposite side there is a ramp with a maximum slope of 8% that meets the regulatory requirements for the elimination of architectural barriers and allows access to the quay with an easier and more attractive route (Figure 22).

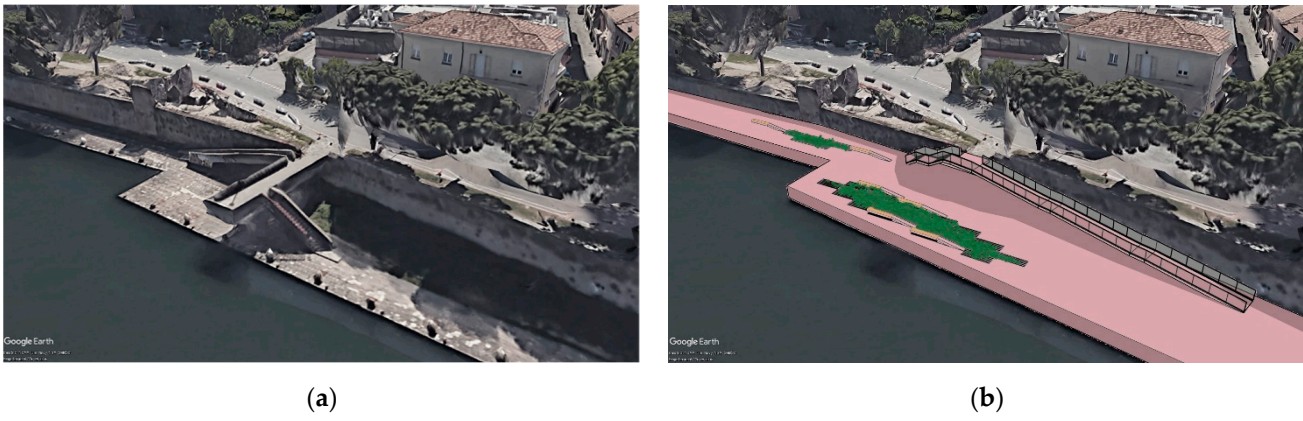

(**a**)          (**b**)

**Figure 22.** (**a**) Before, and (**b**) after dock elevation at Access 9. The pink-highlighted area shows the portion of quay subject to superelevation (© 2022, R. Corticelli—CIRI Building and Construction).

### 4.3.10. Access 10

As the staircase is part of the 1977 renovation by Vittoriano Viganò, here, it is proposed to maintain and redesign it. In this case, there would be no space for the construction of a sloping ramp in accordance with updated standards, and the double sail of Viganò's work can be maintained as a testimonial work of architectural brutalism. The proposal is to upgrade the existing staircase, adapting it to the new elevation of the quay. The only ramp to be maintained is the one that descends towards the Ponte dei Mille, which would become shorter and allow the insertion of an intermediate resting landing. The ramp that currently descends towards the mobile footbridge and Tiberius Bridge would be closed to allow for the widening of the cycle/pedestrian passage on Via Bastioni Settentrionali. By leaving the shape of the sail on the quay, but preventing access to it, since it would be a space that is not very visible and potentially susceptible to social degradation, it is proposed to enclose it with a dense dark steel grating that can be inspected for maintenance purposes (Figure 23).

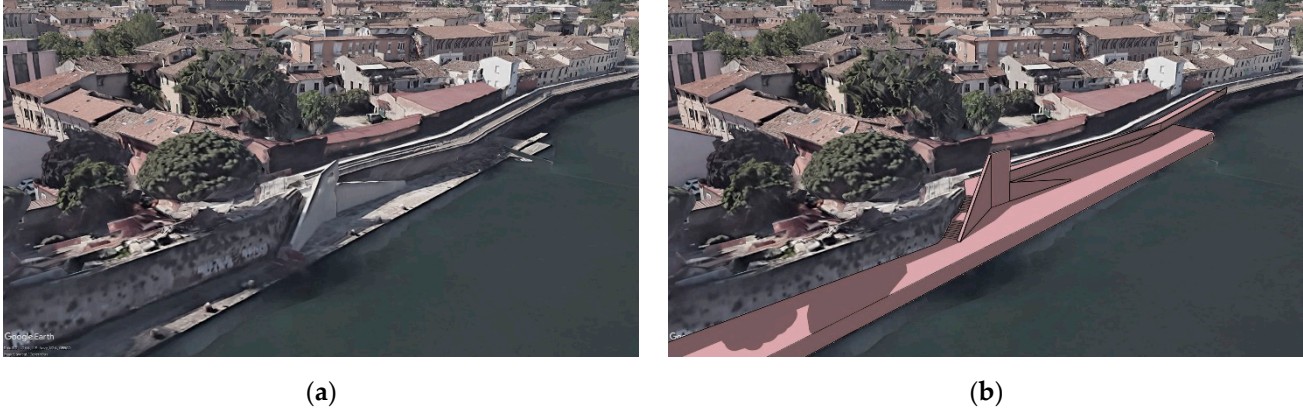

(**a**)          (**b**)

**Figure 23.** (**a**) Before, and (**b**) after dock elevation at Access 10. The pink-highlighted area shows the portion of quay subject to superelevation (© 2022, R. Corticelli—CIRI Building and Construction).

### *4.4. Cycle Paths*

After the study and design of each access point to the quays, a particular focus was given to the study of cycle paths. Although for the most part they were kept above the ancient historic walls, beside the roadway, a dedicated cycle path along Via Marecchia was not proposed on the left side of the canal, as initially assumed. The reason for this choice lies in the fact that a narrowing occurs between Via Marecchia and Via Madonna della Scala, due to the presence of an ancient city gate. In this case, it is proposed to provide promiscuous bicycle and pedestrian paths in the section of the quay between Don Luigi

Sturzo Gardens and Via Madonna della Scala. From here, the new section of the cycle path proposed in the project along Via Sinistra del Porto is connected to this point.

　　The cycle paths in the project area are therefore classified as follows (Figure 24):

- In green, the existing cycle path along Via Destra del Porto, part of the so-called "green ring".
- In yellow, the 30 km/h zone in Borgo San Giuliano.
- In blue, the new cycle/pedestrian path on the regenerated quay.
- In red, the new cycle path from Via Madonna della Scala to Via Sinistra del Porto.

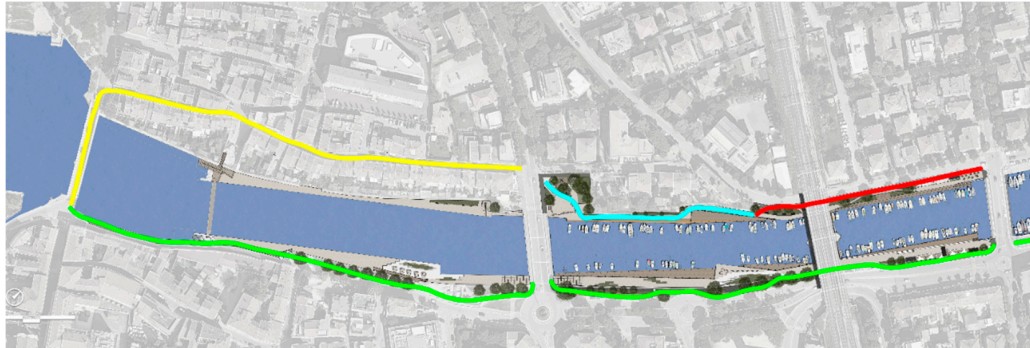

**Figure 24.** Cycle paths in the project area (© 2022, R. Corticelli—CIRI Building and Construction).

　　The existing cycle path along Via Destra del Porto (green line) is properly delimited with specific vertical and horizontal signage and red paving, so no changes are planned in this area in the new configuration. The only issue that has emerged is at Via Bastioni Settentrionali, where there is a bottleneck at the beginning of the cycle/pedestrian path. The closure of the ramp that currently descends towards the moving footbridge and the Tiberius Bridge would allow the widening of the cycle/pedestrian passage on Via Bastioni Settentrionali (see Access 10). Once at the Tiberius Bridge, the cycle path continues beyond the Canal Port along Bicipolitana line 3 (yellow line). Crossing the bridge, one arrives along the left bank of the canal where cyclists can circulate without danger within zone 30 km/h of Borgo San Giuliano. The new square proposed for Access 3 would then allow the cycle route within Borgo San Giuliano to be channelled naturally towards the quaysides, resulting in greater use of these spaces and a better urban quality (blue line). The connection of the cycle path on the quays and Via Sinistra del Porto is possible thanks to the proposed Access 4 of a new cycle/pedestrian passage. Along the left bank of the Port Canal, there is currently no cycle path. The new configuration envisages the construction of a two-way cycle track on the pavement (red line).

　　The width of the new bicycle section is in line with the "Guidelines for the regional bicycle system" of the Emilia–Romagna Region [49]. For bi-directional cycle paths on Category B pavement (secondary network), the sum of the minimum width of the effectively transitable platform (rolling surface), the additional space to be guaranteed with respect to the edges, and the clearance to be guaranteed with respect to lateral obstacles, continuous or discontinuous, must be equal to 2.50 m (regardless of the type of road travelled, the type of separation and the number of cyclists expected). In particular, the transitable surface is the one directly affected by the wheels of the bicycle, which must present the necessary characteristics of regularity, smoothness, and load-bearing capacity; the free space from the edges is an additional empty space to the previous one, which may not be "perfectly transitable" (it may, for example, contain gullies or drains), separating the rolling surface from the edges of the track. The francs to be guaranteed in relation to lateral obstacles must be calculated from the limit of the running surface.

　　For the choice of materials to be used in the different layers of the cycle paving, reference was made to the "Guidelines for the design of interventions on roads, squares and related infrastructures" of the Municipality of Bologna [50].

*4.5. Masterplan*

The output of the preliminary design phase resulted in a master plan that embodies all the design intentions outlined in the previous sessions.

The main intention of reconnecting the cycle and pedestrian routes was the first element that was considered in the design phase, and for this reason, special attention was paid to realising connections that would encourage end-users to converge as much as possible on the quaysides. An increased flow of visitors contributes to the attractiveness of the area and creates a sense of identity for the local community [48]. The new routes were proposed mainly by means of sloping ramps, and where there was no space for a ramp, the stairways were interrupted with rest landings (Figure 25). The new cycle path reflects the guidelines of local regulations, and is designed with permeable materials similar to those used in the creation of the Parco del Mare. The raised docks were designed with the intention of integrating green and permeable areas within them as much as possible. Thanks to the elevation above the current level, it is possible to integrate the flowerbeds directly into the docks' thickness. In addition, as many trees and other types of vegetation as possible were inserted to mitigate the heat island effect and contribute to a reduction in atmospheric pollutants from vehicle and marine traffic. The new flowerbeds also contributed to improving the urban quality of the regenerated spaces, and influence the aesthetic and perceptual appeal of urban spaces. This aesthetic value of urban spaces is very relevant to the success of an urban regeneration action. Indeed, the perception of aesthetically pleasing spaces also contributes to an increased sense of belonging on the behalf of the local community and improves attractiveness for visitors [51].

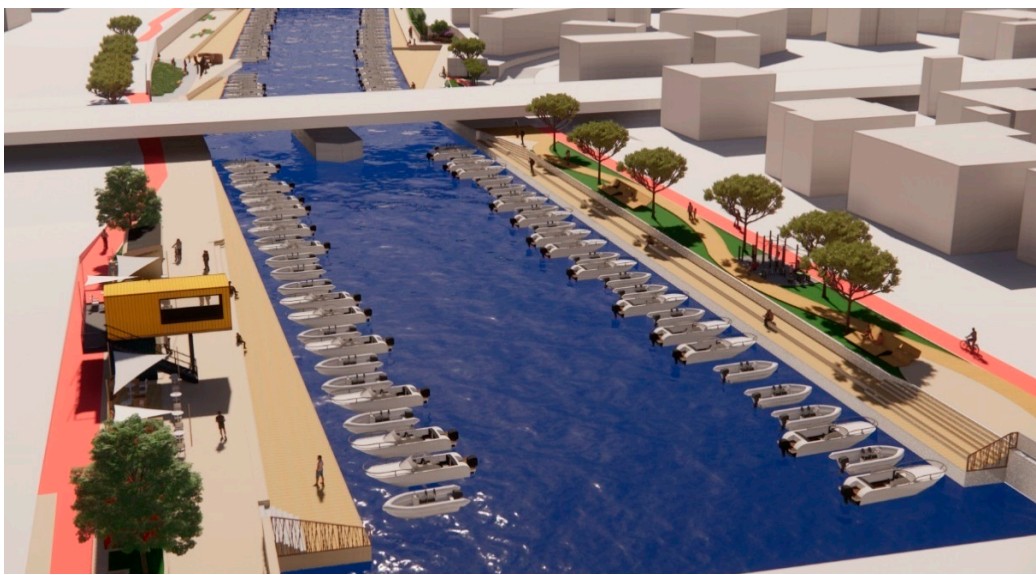

**Figure 25.** Canal Port view from Ponte della Resistenza (© 2022, R. Corticelli—CIRI Building and Construction).

In addition to green spaces, an attempt was made to propose the use of materials with low environmental impact, similar to those used for the urban regeneration of the Parco del Mare and the Piazza sull'Acqua. In addition, the insertion of photovoltaic panels was proposed in various positions: on the Railway Bridge parapet, on the parking and charging shelters for electric bicycles, and on the exhibition panels (no. 4, Figure 26).

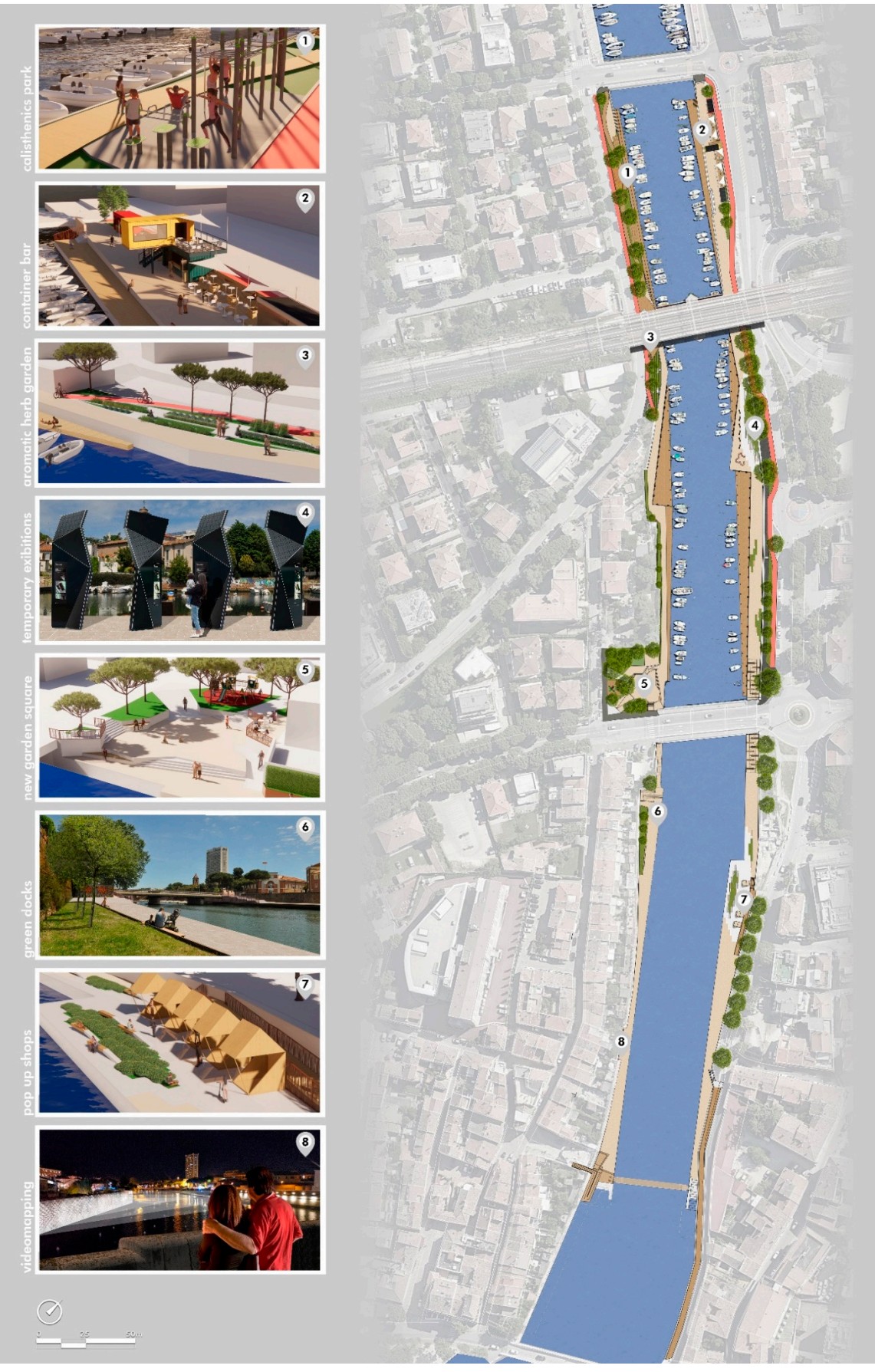

**Figure 26.** Masterplan and public space location (© 2022, R. Corticelli—CIRI Building and Construction).

Finally, in order to make the area even more attractive, various social functions were proposed for the public spaces: leisure and sports facilities (callisthenics park, and playground); commercial spaces (container bars, and pop-up shops); green gathering areas (new square garden, green docks, and aromatic herb garden); and art installations (video mapping installation, and temporary exhibition spaces).

## 5. Discussion and Concluding Remarks

The analysis of the historical and urban context of Rimini Canal Port led to the conclusion that the port area of Rimini is currently poorly valued and poorly connected to the existing urban context. The project presented in this paper aims to create an area that integrates perfectly with the urban context in which it is inserted and highlights the characteristics of historical value that are currently undervalued.

The data gathered from the documentary analysis, as well as from the on-site surveys and consultation with stakeholders, were processed and their evaluation resulted in:

- A SWOT matrix (Strengths—Weaknesses—Opportunities—Threats) based on the results of the preliminary analysis delivered in the first phase;
- in-depth analysis of criticalities and potentials identified by the dialogue with involved stakeholders;
- the creation of a set of significant indicators for the assessment of the urban and infrastructural quality of the Canal Port area;
- BOCR analysis (Benefits—Opportunities—Costs—Risks) based on the selected set of indicators and identification of the priority scale of the interventions to be carried out for the redevelopment of the Canal Port area.

The outcomes of the preliminary analysis phase concluded that the priority actions needing to be carried out were the improvement of cycle/pedestrian paths, the redevelopment and raising of the docks, and the regularisation of the moorings.

This legitimised the project proposal, which was already intended to focus on the implementation of soft mobility as a means of achieving a holistic improvement of the urban quality of the Rimini Canal Port area.

The design phase began with the identification of the height to which to raise the docks in order to solve the main problem of frequent flooding due to tides and adverse weather conditions. Following an in-depth hydraulic study of the area and meetings with stakeholders, a raising of the quays to a height of 1.50 a.s.l. was justified and verified. Downstream of the Ponte della Resistenza Bridge, the quays have already been recently renewed by the Municipality of Rimini. In that section, the docks are never lower than 1.20 m a.s.l., and are not normally submerged as is the case with the section upstream of the Ponte della Resistenza Bridge. For this reason, the design proposal focused on the quays between the Tiberius Bridge and the Ponte della Resistenza Bridge (Figure 27).

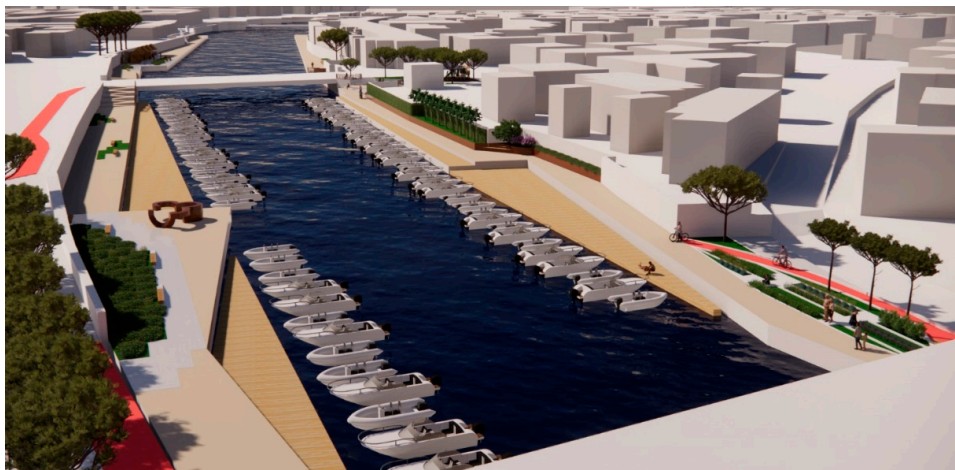

**Figure 27.** Canal Port view from Railway Bridge (© 2022, R. Corticelli—CIRI Building and Construction).

The accesses to the quays and public spaces were studied in order to identify new functions for the benefit of the community. Following the raising of the docks, the bicycle and pedestrian paths along the two banks of the Canal Port were reviewed. The new cycle and pedestrian infrastructure can improve public health and make cities more active and environmentally friendly. Recent studies have shown that the regeneration of urban public spaces is closely related to the presence of safe and connected cycling and pedestrian path [52–55]. The proposed solutions are currently being defined and refined and could receive funding from the Municipality of Rimini to be implemented.

The solutions proposed in this contribution represent targeted and specific interventions that are designed precisely for the context to which they are dedicated, in order to make the Canal Port area a continuum with its urban context and to improve its perception by tourists and inhabitants. Although the proposed solution is tailor-made for this specific case, the approach that has been developed is based on a strong scientific basis of urban regeneration projects founded on multi-criteria analysis and sets of indicators. The applied methodology can be replicated in any other similar case requiring an urban regeneration intervention.

The benefits of this urban regeneration project include:

- improved aesthetic quality of urban spaces;
- improved environmental quality of urban spaces;
- reduction of pollutant emissions through the increase in green and permeable areas;
- increased user flow (residents and tourists) in areas that are currently poorly frequented;
- increased social well-being of the regenerated area.

To verify the effectiveness of urban regeneration actions, the same indicators used in the planning phase can be re-used in the monitoring phase to verify changes compared to the current situation and to confirm the validity of the design choices made.

All these aspects could potentially result in higher economic productivity in the area. Better urban quality may lead to an increase in the real estate value of the area and an implementation of economic activities.

As a future development of the research, it is proposed to deepen an economic feasibility study of the interventions, which in any case should be convenient for the municipality, as it does not involve actions of deep urban transformation and demolition, but simple local interventions of renovation of public spaces.

The proposed urban regeneration project, focused exclusively on the redevelopment of bicycle and pedestrian routes, aims to represent a good example of how soft mobility plays a fundamental role in urban regeneration.

**Author Contributions:** Conceptualization, R.C.; methodology, R.C.; formal analysis, investigation, data curation, R.C. and M.P.; writing—original draft preparation, R.C.; writing—review and editing, R.C., M.P., C.M., C.L., A.F. and V.V.; visualization, R.C., M.P. and C.M.; supervision, R.C., M.P., C.M., C.L., A.F. and V.V. All authors have read and agreed to the published version of the manuscript.

**Funding:** The research was carried out within the "FRAMESPORT" project, funded by the European Interreg Italy–Croatia project under Application ID No 10253074.

**Acknowledgments:** The set of urban regeneration indicators presented was developed in the framework of Chiara Casamassima master's thesis entitled "Strumenti e metodi di analisi e progettazione per la rigenerazione urbana delle aree portuali: il caso del Porto Canale di Rimini" (Alma Mater Studiorum—University of Bologna; supervisor: Ferrante, A.; co-supervisor: Mazzoli, C., Corticelli, R., and Lantieri, C.). The authors would like to thank Fondazione ITL (Istituto sui Trasporti e la Logistica) for providing such an interesting and complex case study to be analysed within the FRAMESPORT project. A special thanks to all the stakeholders involved in the preliminary design phases, and in particular the Director of Infrastructure, Mobility and Environmental Quality of the Municipality of Rimini Eng. Alberto Dellavalle, for having constantly dialogued with UNIBO and ITL in order to develop the project.

**Conflicts of Interest:** The authors declare no conflict of interest. The funders had no role in the design of the study; in the collection, analyses, or interpretation of data; in the writing of the manuscript; or in the decision to publish the results.

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
