# Peer review of "Urban Regeneration and Soft Mobility: The Case Study of the Rimini Canal Port in Italy"

_sustainability, doi:10.3390/su142114529_

Round 1

Reviewer 1 Report

It was with great curiosity and interest that I read the article “Urban Regeneration and Soft Mobility: the case study of the Rimini Canal Port in Italy”. The subject of the study is interesting, pertinent and in line with current trends and research needs about sustainable cities and the importance of soft mobility to achieve that.

This article complies with almost all the established norms for the articles of this publication in organization and offers interesting results. Congratulations about the pictures, it really helps the reader to better understand the topic.

However, I want to suggest some minor revisions to improve the manuscript:

1 – In the Abstract: the methodology used to achieve the proposed objectives are not properly presented. The authors should specify all the methodologies used. For example, it is just along the text that I found that a questionnaire was applied.

 2 – In point 2: Section too long and too descriptive. I understand the relevance of the information but it can be summarized.

3 – In point 3: About the questionnaires, who were the stakeholders? How were they selected? How many were there? When were the questionnaires was carried out? What techniques were used for data analysis? The authors should clarify the methodology process. I had difficult to find the SWOT analysis! This part should be better organized.

4 – Pease check some references because it seems to me that don’t follow the rules of the journal.

5 – In general terms, the article seems too long. I believe that author(s) should make every effort to concentrate the most relevant contribution of their research in fewer pages.

Author Response

REVIEWER#1

It was with great curiosity and interest that I read the article “Urban Regeneration and Soft Mobility: the case study of the Rimini Canal Port in Italy”. The subject of the study is interesting, pertinent and in line with current trends and research needs about sustainable cities and the importance of soft mobility to achieve that.

This article complies with almost all the established norms for the articles of this publication in organization and offers interesting results. Congratulations about the pictures, it really helps the reader to better understand the topic.

  1. In the Abstract: the methodology used to achieve the proposed objectives are not properly presented. The authors should specify all the methodologies used. For example, it is just along the text that I found that a questionnaire was applied.

The authors implemented the suggested reference, increasing the information about the questionnaire in the abstract.

  1. In point 2: Section too long and too descriptive. I understand the relevance of the information but it can be summarized.

The authors reduced paragraph 2.3 (lines 192-258).

  1. In point 3: About the questionnaires, who were the stakeholders? How were they selected? How many were there? When were the questionnaires was carried out? What techniques were used for data analysis? The authors should clarify the methodology process. I had difficult to find the SWOT analysis! This part should be better organized.

The authors thank you for the suggestion. In section 3.2 (lines 298-308) more information about the questionnaire was added. The main stakeholders, to whom the online questionnaire was sent, and the structure were highlighted. More information about the methodology will be found in the article "Multi-criteria analysis and decision-making approach for the urban regeneration: the application to the Rimini Canal Port (Italy)" (under publication). Given the complexity and length of the project, the authors drafted article in which all the phases of the methodology are described in detail.

  1. Please check some references because it seems to me that don’t follow the rules of the journal.

The authors thank you for having noted and pointed it out. The references have been modified according to the rules of the journal.

  1. In general terms, the article seems too long. I believe that author(s) should make every effort to concentrate the most relevant contribution of their research in fewer pages.

The authors thank you for the suggestion. The paper has been revised and shortened accordingly.

Reviewer 2 Report

Dear Authors, 

The article, titled "Urban regeneration and soft mobility: a case study of the Rimini Canal Port in Italy" aims to show the strong impact of soft mobility in urban regeneration projects and how improving the quality of bicycle and pedestrian paths can enhance the quality of urban spaces. 

After reading the paper, I have comments and suggestions to improve the paper as follows: 

Keywords: it is too long. I suggest improving it according to the journal's guidelines 

Introduction 

The research problem is interesting and well presented. The authors have used a rich literature. However, I suggest dividing this chapter into two parts and introducing a Literature Review chapter.  

The Result chapter is very interesting. I suggest describing exactly, what criteria were taken into account in the SWOT analysis? 

In the Discussion section, I suggest explaining in more detail how improving the quality of bicycle and pedestrian paths can improve the quality of urban spaces. It would also be useful to include the results of studies by other European authors.  

The literature list needs to be improved according to the journal's guidelines. 

In conclusion, I recommend this paper for publication in the journal Sustainability after minor significant changes.  

Kind regards,   

Reviewer

Author Response

REVIEWER#2

The article, titled "Urban regeneration and soft mobility: a case study of the Rimini Canal Port in Italy" aims to show the strong impact of soft mobility in urban regeneration projects and how improving the quality of bicycle and pedestrian paths can enhance the quality of urban spaces.

After reading the paper, I have comments and suggestions to improve the paper as follows:

  1. Keywords: it is too long. I suggest improving it according to the journal's guidelines

The authors revises the keywords and shortened them accordingly.

Introduction

  1. The research problem is interesting and well presented. The authors have used a rich literature. However, I suggest dividing this chapter into two parts and introducing a Literature Review chapter.

The authors thank you for the suggestion. The organization of the paragraphs was chosen following the template of the journal in which the Literature Review is within the introduction paragraph.

  1. The Result chapter is very interesting. I suggest describing exactly, what criteria were taken into account in the SWOT analysis?

The authors thank you for the suggestion. In this article, the authors focused on the description of the project, including only a brief introduction to the methodology. Given the complexity and length of the latter, another article was written in which all the phases of the decision-making process (urban indicators, SWOT, and BOCR) were described in detail. Title of the article: "Multi-criteria analysis and decision-making approach for the urban regeneration: the application to the Rimini Canal Port (Italy)" (under publication).

  1. In the Discussion section, I suggest explaining in more detail how improving the quality of bicycle and pedestrian paths can improve the quality of urban spaces. It would also be useful to include the results of studies by other European authors.

The authors added more details about the improvement of the quality of urban spaces through the new cycle-pedestrian paths. Examples from other studies in the literature have been included. (lines 888-892)

  1. The literature list needs to be improved according to the journal's guidelines.

The authors thank you for having noted and pointed it out. The references have been modified following the guidelines of the journal.

In conclusion, I recommend this paper for publication in the journal Sustainability after minor significant changes. 
